# Learning to Generate Predictor for Long-Term Time Series Forecasting

## Abstract

Long-term time series forecasting (LTSF) is a significant challenge in machine learning with numerous real-world applications. Although transformer architecture have shown promising performance in the LTSF task, recent research suggests that they are not suitable for time series forecasting due to their permutation invariant characteristic, and proposes a simple linear predictor which outperforms all existing transformer architectures. However, the linear predictor is inflexible and cannot reflect the characteristics of the time series for prediction due to its simple architecture. In this paper, we introduce a novel Learning to Generate Predictor (LGPred) framework, which generates a linear predictor adaptively to the given input time series by leveraging time series decomposition. LGPred obtains representations from the decomposed time series and generates a predictor suitable for the given time series from these representations. Our extensive experiments demonstrate that LGPred achieves state-of-the-art performance for both multivariate and univariate forecasting tasks.

## 1 Introduction

Forecasting the future is one of the most important goal of the artificial intelligence. Time series forecasting (TSF) has been extensively studied as a means of forecasting the future by using past values to predict future values of a time series. The prediction capability of TSF models can be applied to diverse real-world applications, such as weather forecasting, traffic control, energy management, and financial investment. Among the time series forecasting methods, transformer-based approaches have shown great successes in more challenging long-term time series forecasting (LTSF) problem. Transformer (Vaswani et al., 2017) models have achieved great success in many areas such as natural language processing (Devlin et al., 2018) and computer vision (Dosovitskiy et al., 2020), demonstrating remarkable ability in capturing long-term dependencies in sequential data. Inspired by the success of transformer models on sequential data, there have been many approaches to utilize transformer architecture for the LTSF problem (Wu et al., 2021; Zhou et al., 2021; 2022). Among them, Informer (Zhou et al., 2021) focuses on reducing computation cost of transformer to handle long time series. On the other hand, Autoformer (Wu et al., 2021) and Fedformer (Zhou et al., 2022) try to apply time series decomposition to better capture the characteristics of the time series.

Despite of its remarkable achievements on LTSF problem, the effectiveness of the transformer architecture for LTSF remains controversial. A recent study (Zeng et al., 2023) argues that the self-attention mechanism of transformer is unsuitable for time series forecasting due to its permutation-invariant characteristics. Since each timestamp of time series contains only a small amount of semantic information, it is crucial to capture temporal relationship among the consecutive timestamps for time series forecasting, which is a lacking ability in self attention. In the paper, the authors propose a simple baseline utilizing only a single linear layer as a predictor and show that the proposed linear model outperforms the complicated transformer-based predictors. Nevertheless, the linear predictor is also not an optimal solution. Due to its simplistic architecture, it is difficult for the linear predictor to capture the characteristics of a dynamically changing time series and reflect them in prediction. While the characteristics of the time series vary dynamically across different time points, the linear model exclusively encodes the temporal dependencies commonly shared across the entire time series within the fixed parameters of the linear layer, failing to capture the distinct characteristics of each input time series individually. We experimentally confirm that the limitation

of linear predictor actually exists and the linear predictor can not learn to predict the time series with diverse trends and seasonalities.

To address the limitation of linear predictor, we propose a new framework called Learning to Generate Predictor (LGPred) which enables the dynamic adaptation for the linear predictor to a given time series by adaptively generating a linear predictor based on the characteristics of time series. Learning to generate model parameters is a useful technique that allows a model to adapt itself to a given task, proven to be effective in fields such as meta learning (Munkhdalai & Yu, 2017; Rusu et al.) or language modeling (Üstün et al., 2020; Mahabadi et al., 2021). Inspired by meta-learning models, each time series is treated as an individual task, allowing the model to adapt to the specific characteristics of each time series for more accurate predictions within our LGPred framework. LGPred adopts time series decomposition to better capture and utilize the characteristics of the time series, dividing it into trend and seasonality components. We extract features through an architecture suited to each component and generate the predictor based on these features. Furthermore, to leverage shared information across various time series and efficiently generate the predictor, we use a template predictor trained across multiple time series with a bottleneck architecture, generating only the necessary part of the linear predictor. Our extensive experiments demonstrate that LGPred achieves state-of-the-art performance on six benchmarks. The contributions of the paper are summarized as follows:

- We introduce LGPred which generates the linear predictor adaptively to the given time series to overcome the limitation of linear time series predictor which fail to predict time series with varying trend and seasonality characteristics.

- To the best of our knowledge, our LGPred is the first attempt at adaptively generating a predictor reflecting the characteristics of the each time series. This approach enables the explicit adaptation of the model to characteristics of the each time series. We believe that our methodology for adaptively generating predictors can be extended to various architectures for time series prediction, further advancing the prediction capability.

- Our empirical studies show that proposed LGPred achieves the state-of-the-art performance on six different benchmarks covering various real-world domains of disease, economics, energy, traffic, and weather.

## 2 PRELIMINARY

In this section, we define the long-term time series forecasting (LTSF) problem and provide the necessary notation to describe it. Given the time series with the number of features or channels $m$, let $\mathbf{x}_t \in \mathbb{R}^m$ as the $t$-th timestamp of a time series. The objective of time series forecasting is to predict future values of time series $\mathbf{X}_{t:t+H} = [\mathbf{x}_t, \cdots, \mathbf{x}_{t+H-1}] \in \mathbb{R}^{H \times m}$ given the look-back window $\mathbf{X}_{t-L:t} = [\mathbf{x}_{t-L}, \cdots, \mathbf{x}_{t-1}] \in \mathbb{R}^{L \times m}$ where $L$ is the look-back window size and $H$ is the length of forecasting horizon. When the number of variates $m = 1$, we refer the problem as a univariate time series forecasting. On the other hand, when $m > 1$, we denote the problem as a multivariate time series forecasting. LTSF refers the special case of time series forecasting with large $T$. In general, time series forecasting with $T > 48$ is considered as LTSF. For simplicity, we denote lookback window $\mathbf{X}_{t-L:t}$ and forecasting horizon $\mathbf{X}_{t:t+H}$ as $\mathcal{X}$ and $\mathcal{Y}$ in the following sections.

## 3 METHOD

In this section, we provide a detailed explanation of the LGPred model. The architecture of the proposed LGPred framework is illustrated in Figure 1. LGPred is designed to identify the underlying characteristics of a given time series and generate an appropriate predictor. We use a time series decomposition technique and extract representations using networks tailored for each decomposed component to better capture the unique characteristics of each time series. Using the extracted representations of all components,the predictor suitable for a given time series is derived from the predictor generator. The generated predictor is applied independently to all channels of the time series, without considering any interdependence between them, similar to the linear predictor proposed in (Zeng et al., 2023). To leverage common knowledge that can be shared for all time series, we utilize a template predictor that is trained across various time series, with the generated predictor reflecting the specific characteristics of the given time series. To cope with distribution shift and further enhance

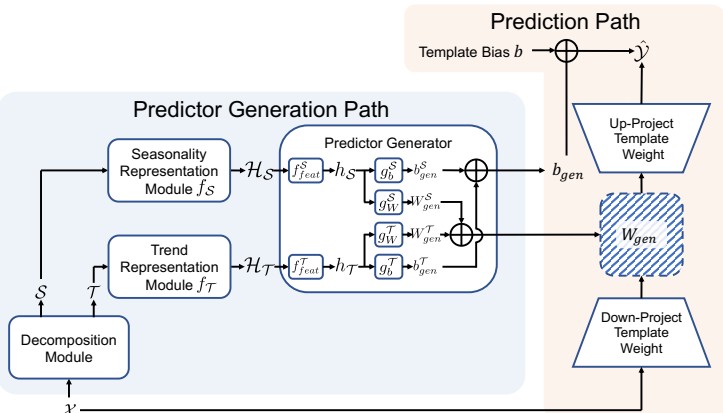

Figure 1: LGPred architecture overview. LGPred consists of decomposition module, representation modules, predictor generator, and template predictor. The decomposition module decomposes the input time series into trend and seasonality. Representation modules extract the corresponding representation from each trend and seasonality. The predictor generator generates weight and bias from the representations. Template predictor leverages the common knowledge shared across all time series, and a weight generated by the predictor generator is applied between the down-project and up-project weights of the template predictor.

the performance, we employ a normalization technique that subtracts the last value of the input from the input time series and adds it to the output prediction, as proposed in (Zeng et al., 2023). Note that we only apply normalization only for prediction, and using the original time series for predictor generation. LGPred is trained using mean-squared error (MSE).

**Decomposition module** We use time series decomposition, a standard technique for time series analysis (Cleveland et al., 1990), to easily identify the characteristics of the time series. Through this process, a complex time series can be decomposed into more understandable components. We adopt the commonly used time series decomposition scheme found in various prior works (Wu et al., 2021; Zeng et al., 2023; Zhou et al., 2022). Specifically, we first decompose the trend $\mathcal{T}$ component using a moving average kernel with the padding, and then use the remainder obtained by subtracting the trend from the original series as the seasonality component $\mathcal{S}$.

**Representation module** To extract representations from the trend and seasonality, we utilize networks specifically designed to capture the characteristics of each component. Our design objective is to employ a network that is tailored to the specific attributes of each component. Given that a trend component encompasses long-term tendencies and a seasonality component encompasses repetitive short-term patterns, a network dedicated to a trend should be capable of capturing global long-range dependencies, while a network for a seasonality should emphasize short-term local dependencies. For the trend component, we firstly explore three architectures that have the capability to capture global dependencies: fully connected network, recurrent neural network (RNN), and transformer. However, transformers are not well-suited for modeling sequential relationships in time series (Zeng et al., 2023), and RNNs have limitations in effectively handling long-term dependencies. As a result, we use a fully-connected network as a representation extraction module for trend component. We believe that a fully-connected layer that allows for interaction between all timestamps is appropriate for extracting representation from the trend component. However for multivariate time series, we also need to consider the interaction between different features. To address the issue of interaction between different features in multivariate time series, we draw inspiration from MLP-mixer (Tolstikhin et al., 2021) and introduce another fully-connected layer that operates on the feature dimension. The left side of Figure. 2 depicts the network architecture of the fully-connected block with our time and channel fully-connected layers.

For the seasonality component, we employ a convolutional neural network which is designed to capture the local patterns from the data by exploiting the locality properties of data (Krizhevsky et al., 2017). Since the seasonality exhibits locality by containing repeated short-term patterns, the temporal convolutional network is well-suited for effectively capturing the characteristics of the seasonality component. Our assumption is that the temporal convolution network using the filter shared across

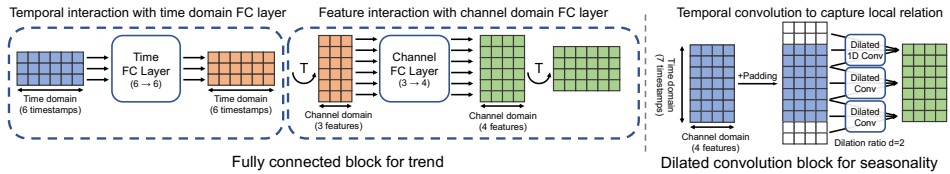

Figure 2: Network architecture of building blocks of trend (left) and seasonality (right) representation module. The trend representation module uses fully-connected layers as building blocks, operating in both the time and feature dimensions. The seasonality representation module uses a dilated convolution layer as a building block. Both building blocks use the GeLU activation function to introduce nonlinearity in each layer.

the time windows is useful for capturing the seasonal component, where similar patterns appear repeatedly. We utilize a dilated temporal convolutional network that introduces gaps between kernel elements of temporal convolutional networks shown on the right side of Figure. 2 to extract the representation. Dilated convolution helps our seasonality module to capture the patterns with various lengths. Note that for both trend and seasonality, our building blocks do not change the length of feature; Only the number of channels changes as the feature passes through a block. The work of representation modules can be expressed by:

$$\mathcal{H}_\mathcal{T} = f_\mathcal{T}(\mathcal{T}) \qquad \mathcal{H}_\mathcal{S} = f_\mathcal{S}(\mathcal{S}),$$

where $f_\mathcal{T}$ denotes the network consists of fully-connected layers and $f_\mathcal{S}$ denotes the network composed by dilated convolution layers with increasing dilation ratios. The extracted representations $\mathcal{H}_\mathcal{T}$ and $\mathcal{H}_\mathcal{S}$ are in $\mathbb{R}^{L \times d_{rep}}$, where $d_{rep}$ is the dimension of representation.

**Predictor generator** We use the trend and seasonality representations, $\mathcal{H}_\mathcal{T}$ and $\mathcal{H}_\mathcal{S}$, to compute the weight and bias of the predictor. To generate these parameters, we first flatten $\mathcal{H}_\mathcal{T}$ and $\mathcal{H}_\mathcal{S}$ into vectors with a length of $L \times d_{rep}$. Since $L \times d_{rep}$ is usually very large in LTSF situation, we compress the representations into the feature $h_\mathcal{T}$ and $h_\mathcal{S}$ with a length of $d_{feat}$.

$$h_\mathcal{T} = f_{feat}^\mathcal{T}(\mathcal{H}_\mathcal{T}) \qquad h_\mathcal{S} = f_{feat}^\mathcal{S}(\mathcal{H}_\mathcal{S}).$$

Here, $f_{feat}^\mathcal{T}$ and $f_{feat}^\mathcal{S}$ are fully-connected layers with GeLU activation. Using the compressed features, we generate the weights and biases with the linear layers. The process can be displayed as:

$$W_{gen}^\mathcal{T} = g_W^\mathcal{T}(h_\mathcal{T}) \qquad b_{gen}^\mathcal{T} = g_b^\mathcal{T}(h_\mathcal{T})$$
$$W_{gen}^\mathcal{S} = g_W^\mathcal{S}(h_\mathcal{S}) \qquad b_{gen}^\mathcal{S} = g_b^\mathcal{S}(h_\mathcal{S}),$$

where $g_W^\mathcal{T}$ and $g_W^\mathcal{S}$ are linear layers whose output size is the number of parameters of the weight, while $g_b^\mathcal{T}$ and $g_b^\mathcal{S}$ are linear layers with an output size of $H$ for the bias. The final weight and bias are obtained by summing the weights and biases generated from the trend and seasonality, respectively.

$$W_{gen} = W_{gen}^\mathcal{T} + W_{gen}^\mathcal{S} \qquad b_{gen} = b_{gen}^\mathcal{T} + b_{gen}^\mathcal{S}.$$

**Template predictor** Although the predictor generator in LGPred aims to leverage the specific characteristics of a given time series, it is important to consider the commonalities that many time series share. Therefore, we introduce a template predictor that is trained across the all time series in the training set. Inspired by the adapter (Houlsby et al., 2019), we utilize the template predictor in the form of a bottleneck architecture. Within the bottleneck architecture, we employ two template predictor weights: down-project weight that projects the input series with length $L$ into latent feature with a smaller dimension of $d_{latent}$, and up-project weight which predicts the output series of length $H$ based on the latent feature. The weight $W_{gen}$ from the weight generator is applied to the latent feature obtained from the down-project weight, and the result is subsequently passed to the up-project template weight. While $W_{gen}$ is generated reflecting the characteristics of each specific time series, the up-project and down-project template weights learn the common characteristics shared by various time series. By utilizing the template predictor, we can leverage the knowledge shared across various time series instead of generating a predictor containing the same information every time.

Another advantage of our template predictor is that it prevents the huge computational and memory cost due to generate the entire predictor weight directly. In the case of directly generating the weight of

size $L \times H$, the scalability of the predictor generation is usually limited by inefficiently huge number of weight parameter, which involves training two linear weights $g_W^{\mathcal{T}}$ and $g_W^{\mathcal{S}}$ with approximately 133M parameters when $L = 720$, $H = 720$, and $d_{feat} = 256$, given by $L \times H \times d_{feat}$. When the template predictor with the bottleneck architecture is adopted, on the other hand, we only generate $W_{gen}$, a square matrix with dimensions $d_{latent} \times d_{latent}$ instead of generating the entire weight matrix of size $L \times H$. This reduction in the number of generated weight parameters leads to a significant decrease in the cost of weight generation; the number of parameters in the weight generator $g_W$ is reduced from $L \times H \times d_{feat}$ to $d_{latent} \times d_{latent} \times d_{feat}$. For example, if we use $d_{latent} = 128$ in aforementioned $L = 720$, $H = 720$, $d_{feat} = 256$ situation, the number of parameters in weight generator $g_W$ is reduced from 133M to 4M. We incorporate the bias of the template predictor by adding it to the generated bias.

## 4 RELATED WORK

### 4.1 TIME SERIES FORECASTING

Time series forecasting has been an important task of artificial intelligence in that it could assist important decision making processes by predicting future. To solve the TSF problem, various approaches from statistical methods such as ARIMA (Ariyo et al., 2014; Box & Jenkins, 1968; Box & Pierce, 1970) to deep learning methods (Bai et al., 2018; Challu et al., 2023; Lai et al., 2018; Oreshkin et al., 2020; Salinas et al., 2020) have been proposed. Moreover, there have been recent efforts to utilize the transformer architecture (Vaswani et al., 2017) that shows good performances in sequence modeling for time series forecasting. The biggest challenge in applying the transformer to time series forecasting is quadratic computational complexity of self-attention process, and many works have tried to reduce the cost of transformer (Li et al., 2019; Liu et al., 2021; Zhou et al., 2021). Although there have been many studies to utilize deep neural networks and transformer architectures as above, one recent study argues that the transformer architectures are not suitable for time series forecasting due to their permutation invariant characteristic (Zeng et al., 2023). Instead of predictor with deep networks, they propose a simple baseline consists of a single linear layer and show that the linear baseline outperforms all the deep learning based forecasting methods. In contrast to the perspective of Zeng et al. (2023), PatchTST (Nie et al., 2022) presents a transformer-based architecture that surpasses the efficacy of a linear predictor. PatchTST utilizes the temporal information of time series by partitioning it into multiple subseries or patches, which serve as input tokens for the transformer. TimesNet (Wu et al., 2022) divides time series into multiple subseries characterized by varying temporal periods, and utilizes the convolution-based inception blocks to effectively capture relationships both within and between these periods. DeepTime (Woo et al., 2023) embraces the concept of implicit neural representation (INR) and meta learning to time series forecasting. DeepTime introduces a novel differentiable closed form solver to facilitate efficient meta learning for the INR model. TSMixer Chen et al. (2023) utilizes a mixer architecture Tolstikhin et al. (2021) to leverage cross variate correlation in multivariate time series. Inspired by Zeng et al. (Zeng et al., 2023), our LGPred incorporates the linear layer for time series forecasting. However, instead of utilizing the fixed linear predictor for all time series, LGPred adaptively generates the linear predictor tailored for the given time series. Moreover, similar to DeepTime, LGPred incorporates a mechanism for adapting to the given time series. While DeepTime achieves this adaptation through the mathematical resolution of a closed-form equation, our approach, LGPred, employs a distinct network that generates the model parameters reflecting the distinct characteristics of each time series. Our LGPred also bears some resemblence with TSMixer, in terms of utilizing the mixer architecture to capture cross-variate correlation.

Meanwhile, there have been studies to utilize the time series decomposition technique with deep neural networks. Autoformer (Wu et al., 2021) adopts time series decomposition to handle complicated temporal pattern and proposes auto-correlation attention to utilize the repeating patterns for time series forecasting. Fedformer (Zhou et al., 2022) utilizes mixture of experts decomposition to better capture the characteristics of time series, and leverage frequency domain information using Fourier enhanced block or wavelet enhanced block. ETSformer (Woo et al., 2022) integrates a time series decomposition technique of exponential smoothing into the transformer architecture by introducing exponential smoothing attention and frequency attention mechanisms. While these methods adopt the decomposition through moving average kernel, there exists some other methods that decompose the time series by passing through stacked learnable layers. N-BEATS (Oreshkin et al., 2020)

proposes a hierarchical doubly residual stacking method which conducts forecast of future time series and backcast of input time series at the same time. The doubly residual architecture provides hierarchical decomposition based on trained neural basis. Built upon N-BEATS, N-HiTS (Challu et al., 2023) provides an interpretable decomposition via multi-rate data sampling on input time series, and enhances forecasting performance by effectively combining the forecasting outputs from different sampling rates using the hierarchical interpolation. Time series decomposition also produces favorable outcomes when applied to linear predictor. DLinear is a variant of linear predictor applying time series decomposition proposed in (Zeng et al., 2023), and the experiments show that DLinear outperforms the pure linear predictor in most cases. Our LGPred relies on time series decomposition as done in prior works (Wu et al., 2021; Zhou et al., 2022; Zeng et al., 2023). The prior works utilizes the decomposed trend and seasonality directly for time series forecasting. However, we believe the time series decomposition could be more useful in analyzing and understanding the characteristics of time series rather than directly using the decomposed trend and seasonality as input for the forecasting model. Therefore, we only use the decomposed time series to identify the characteristics of the time series and generate a predictor based on them, instead of using them directly for prediction.

## 4.2 LEARNING TO GENERATE NETWORK

Learning to generate the parameters of the network based on the input is the way for the model to obtain good flexibility and adaptability (Denil et al., 2013; Schmidhuber, 1992). Since Schmidhuber (1992) first introduced the idea of learning to generating input-adaptive parameter, this methodology has spread to various tasks (Jia et al., 2016; Ha et al., 2016). For computer vision tasks, DFN (Jia et al., 2016) proposes a filter-generating network which adaptively generates the filters of a convolutional layer. Hypernetwork (Ha et al., 2016) introduces the concept of genotype (the hypernetwork) and phenotype (the main network) utilizing a small hypernetwork to generate parameters of main network. The concept of hypernetwork has applied to various tasks such as visual reasoning (Perez et al., 2018), zero-shot image classification (Jin et al., 2020), and language model (Üstün et al., 2020; Mahabadi et al., 2021). Learning to generate the network parameters is also closely related to meta learning (Munkhdalai & Yu, 2017; Oreshkin et al., 2018; Rusu et al.). These works utilize a few labeled samples for generating the network parameter to adapt to the given task. For example, meta network (Munkhdalai & Yu, 2017) generates the weights using the meta-learner for fast adaptation to the new task. TADAM (Oreshkin et al., 2018) adopts the concept of FiLM (Perez et al., 2018) to generate the conditioning parameters that modifies the feature extractor to obtain a task-adaptive representation. LEO (Rusu et al.) generates the high-dimensional initial parameters of the output linear layer from low dimensional latent feature. While the concept of learning to generate parameters are widely applied to various tasks, our LGPred is the first work to adopt the concept of learning to generate in time series forecasting. Specifically, our LGPred is inspired by Hyperformer (Mahabadi et al., 2021) and Udapter (Üstün et al., 2020) in terms of utilizing the bottleneck architecture, and LEO (Rusu et al.) in terms of generating the high-dimensional parameter from low-dimensional latent.

## 5 EXPERIMENTS

### 5.1 EXPERIMENTAL SETTINGS

**Dataset** For experiments, we use following 6 datasets. 1) Electricity Transformer Temperature (ETT) dataset (Zhou et al., 2021) contains the data from electricity transformer such as load and oil temperature recorded every 15 minutes from July 2016 to July 2018. ETT dataset contains two 15-minute-level datasets (ETTm1, ETTm2) and two hourly-level datasets (ETTh1, ETTh2). 2) Electricity is a dataset of hourly electricity consumption of 321 different clients, collected from 2012 to 2014. 3) Exchange (Lai et al., 2018) collects the daily exchange rates of 8 different countries, from 1990 to 2016. 4) ILI dataset is a weekly record of number of influenza-like illness patients collected by Centers for Disease Control and Prevention of United States, from 2002 to 2021. 5) Traffic dataset is an hourly dataset of the road occupancy rates of 862 different points in San Francisco Bay area freeways, collected by California Department of Transportation from 2015 to 2016. 6) Weather dataset contains 21 meteorological indicators such as air temperature and humidity, recorded every 10 minutes for 2020 whole year. For all datasets, we split the dataset into training, validation, and test set in chronological order, following standard split used in prior works (Wu et al., 2021; Zeng et al., 2023; Zhou et al., 2021; 2022).

**Implementation details** We implement our method using Pytorch, and the experiments are conducted on a single NVIDIA RTX 3090 24GB GPU. We train our LGPred with Mean Squared Error (MSE) loss using Adam optimzier (Kingma & Ba, 2015). See appendix for detailed hyperparameter settings.

**Baselines** We compare our LGPred with six transformer-based methods of Informer (Zhou et al., 2021), Autoformer (Wu et al., 2021), Fedformer (Zhou et al., 2022), ETSformer (Woo et al., 2022), Non-stationary Transformer (Liu et al., 2022), and PatchTST (Nie et al., 2022), the linear predictor combined with decomposition scheme (DLinear) (Zeng et al., 2023), INR-based method of DeepTime (Woo et al., 2023), and convolution-based method of TimesNet (Wu et al., 2022). We evaluate the forecasting models using Mean Squared Error(MSE) and Mean Absolute Error (MAE), following previous works (Wu et al., 2021; Zeng et al., 2023; Zhou et al., 2021; 2022).

## 5.2 RESULTS

Table 1 shows the experiment results on multivariate long-term forecasting. Our LGPred shows the best or the second best MSE and MAE performances in most cases. LGPred achieves the best and second best performance for the MSE metric in 20 and 16 out of 36 experiments, and for the MAE metric in 16 and 16 out of 36 experiments. In Table 2, univariate forecasting experiment results on ETT dataset are displayed. In this experiment, LGPred achieves the best MSE and MAE performances in all experiments except one case on the ETTh2 dataset.

| Method | | LGPred | | PatchTST† | | TimesNet | | DeepTime | | DLinear | | ETSformer | | FEDformer | | Autoformer | | Informer | |
|---|---|---|---|---|---|---|---|---|---|---|---|---|---|---|---|---|---|---|---|
| Metric | | MSE | MAE | MSE | MAE | MSE | MAE | MSE | MAE | MSE | MAE | MSE | MAE | MSE | MAE | MSE | MAE | MSE | MAE |
| Electricity | 96 | **0.129** | 0.229 | **0.129** | **0.222** | 0.168 | 0.272 | 0.137 | 0.238 | 0.140 | 0.237 | 0.187 | 0.304 | 0.193 | 0.308 | 0.201 | 0.317 | 0.274 | 0.368 |
| | 192 | **0.143** | 0.243 | 0.147 | **0.240** | 0.184 | 0.289 | 0.152 | 0.252 | 0.153 | 0.249 | 0.199 | 0.315 | 0.201 | 0.315 | 0.222 | 0.334 | 0.296 | 0.386 |
| | 336 | **0.157** | **0.257** | 0.163 | 0.259 | 0.198 | 0.300 | 0.166 | 0.268 | 0.169 | 0.267 | 0.212 | 0.329 | 0.214 | 0.329 | 0.231 | 0.338 | 0.300 | 0.394 |
| | 720 | **0.189** | **0.290** | 0.197 | **0.290** | 0.220 | 0.320 | 0.201 | 0.302 | 0.203 | 0.301 | 0.233 | 0.345 | 0.246 | 0.355 | 0.254 | 0.361 | 0.373 | 0.439 |
| ETTh1 | 96 | 0.373 | **0.395** | **0.370** | 0.400 | 0.384 | 0.402 | - | - | 0.375 | 0.399 | - | - | 0.376 | 0.419 | 0.449 | 0.459 | 0.865 | 0.713 |
| | 192 | 0.408 | 0.417 | 0.413 | 0.429 | 0.436 | 0.429 | - | - | **0.405** | **0.416** | - | - | 0.420 | 0.448 | 0.500 | 0.482 | 1.008 | 0.792 |
| | 336 | 0.431 | **0.430** | 0.422 | 0.440 | 0.491 | 0.469 | - | - | 0.434 | 0.450 | - | - | 0.459 | 0.465 | 0.521 | 0.496 | 1.107 | 0.809 |
| | 720 | **0.442** | **0.460** | 0.447 | 0.468 | 0.521 | 0.500 | - | - | 0.472 | 0.490 | - | - | 0.506 | 0.507 | 0.514 | 0.512 | 1.181 | 0.865 |
| ETTh2 | 96 | **0.271** | **0.336** | 0.274 | **0.336** | 0.340 | 0.374 | - | - | 0.289 | 0.353 | - | - | 0.346 | 0.388 | 0.358 | 0.397 | 3.755 | 1.525 |
| | 192 | **0.333** | **0.378** | 0.339 | 0.379 | 0.402 | 0.414 | - | - | 0.383 | 0.418 | - | - | 0.429 | 0.439 | 0.456 | 0.452 | 5.602 | 1.931 |
| | 336 | 0.351 | 0.399 | 0.329 | 0.384 | 0.452 | 0.452 | - | - | 0.448 | 0.465 | - | - | 0.496 | 0.487 | 0.482 | 0.486 | 4.721 | 1.835 |
| | 720 | 0.388 | 0.425 | 0.379 | 0.422 | 0.462 | 0.468 | - | - | 0.605 | 0.551 | - | - | 0.463 | 0.474 | 0.515 | 0.511 | 3.647 | 1.625 |
| ETTm1 | 96 | 0.296 | 0.346 | **0.290** | **0.342** | 0.338 | 0.375 | - | - | 0.299 | 0.343 | - | - | 0.379 | 0.419 | 0.505 | 0.475 | 0.672 | 0.571 |
| | 192 | **0.330** | **0.363** | 0.332 | 0.369 | 0.374 | 0.387 | - | - | 0.335 | 0.365 | - | - | 0.426 | 0.441 | 0.553 | 0.496 | 0.795 | 0.669 |
| | 336 | 0.366 | **0.391** | 0.366 | 0.392 | 0.410 | 0.411 | - | - | 0.369 | 0.386 | - | - | 0.445 | 0.459 | 0.621 | 0.537 | 1.212 | 0.871 |
| | 720 | 0.420 | **0.420** | 0.416 | 0.420 | 0.478 | 0.450 | - | - | 0.425 | 0.421 | - | - | 0.543 | 0.490 | 0.671 | 0.561 | 1.166 | 0.823 |
| ETTm2 | 96 | **0.162** | **0.252** | 0.165 | 0.255 | 0.187 | 0.267 | 0.166 | 0.257 | 0.167 | 0.260 | 0.189 | 0.280 | 0.203 | 0.287 | 0.255 | 0.339 | 0.365 | 0.453 |
| | 192 | **0.217** | **0.289** | 0.220 | 0.292 | 0.249 | 0.309 | 0.225 | 0.302 | 0.224 | 0.303 | 0.253 | 0.319 | 0.269 | 0.328 | 0.281 | 0.340 | 0.533 | 0.563 |
| | 336 | **0.274** | **0.327** | 0.274 | 0.329 | 0.321 | 0.351 | 0.277 | 0.336 | 0.281 | 0.342 | 0.314 | 0.357 | 0.325 | 0.366 | 0.339 | 0.372 | 1.363 | 0.887 |
| | 720 | **0.358** | **0.389** | 0.362 | 0.385 | 0.408 | 0.403 | 0.383 | 0.409 | 0.397 | 0.421 | 0.414 | 0.413 | 0.421 | 0.415 | 0.433 | 0.432 | 3.379 | 1.338 |
| Exchange | 96 | **0.078** | **0.195** | - | - | 0.107 | 0.234 | 0.081 | 0.205 | 0.081 | 0.203 | 0.085 | 0.204 | 0.148 | 0.278 | 0.197 | 0.323 | 0.847 | 0.752 |
| | 192 | **0.145** | **0.278** | - | - | 0.226 | 0.344 | 0.151 | 0.284 | 0.157 | 0.293 | 0.182 | 0.303 | 0.271 | 0.380 | 0.300 | 0.369 | 1.204 | 0.895 |
| | 336 | **0.219** | **0.348** | - | - | 0.367 | 0.448 | 0.314 | 0.412 | 0.305 | 0.414 | 0.348 | 0.428 | 0.460 | 0.500 | 0.509 | 0.524 | 1.672 | 1.036 |
| | 720 | **0.426** | **0.515** | - | - | 0.964 | 0.746 | 0.856 | 0.663 | 0.643 | 0.601 | 1.025 | 0.774 | 1.195 | 0.841 | 1.447 | 0.941 | 2.478 | 1.310 |
| ILI | 24 | 1.754 | 0.885 | **1.319** | 0.754 | 2.317 | 0.934 | 2.425 | 1.086 | 2.215 | 1.081 | 2.527 | 1.020 | 3.228 | 1.260 | 3.483 | 1.287 | 5.764 | 1.677 |
| | 36 | 1.702 | 0.835 | **1.430** | 0.834 | 1.972 | 0.920 | 2.231 | 1.008 | 1.963 | 0.963 | 2.615 | 1.007 | 2.679 | 1.080 | 3.103 | 1.148 | 4.755 | 1.467 |
| | 48 | 1.792 | 0.924 | **1.553** | 0.815 | 2.238 | 0.940 | 2.230 | 1.016 | 2.130 | 1.024 | 2.359 | 0.972 | 2.622 | 1.078 | 2.669 | 1.085 | 4.763 | 1.469 |
| | 60 | 1.862 | 0.941 | **1.470** | 0.788 | 2.027 | 0.928 | 2.143 | 0.985 | 2.368 | 1.096 | 2.487 | 1.016 | 2.857 | 1.157 | 2.770 | 1.125 | 5.264 | 1.564 |
| Traffic | 96 | **0.355** | 0.269 | 0.360 | 0.249 | 0.593 | 0.321 | 0.390 | 0.275 | 0.410 | 0.282 | 0.607 | 0.392 | 0.587 | 0.366 | 0.613 | 0.388 | 0.719 | 0.391 |
| | 192 | **0.376** | 0.279 | 0.379 | 0.256 | 0.617 | 0.336 | 0.402 | 0.278 | 0.423 | 0.287 | 0.621 | 0.399 | 0.604 | 0.373 | 0.616 | 0.382 | 0.696 | 0.379 |
| | 336 | 0.393 | 0.292 | 0.392 | 0.264 | 0.629 | 0.336 | 0.415 | 0.288 | 0.436 | 0.296 | 0.622 | 0.396 | 0.621 | 0.383 | 0.622 | 0.337 | 0.777 | 0.420 |
| | 720 | 0.430 | 0.301 | 0.432 | 0.286 | 0.640 | 0.350 | 0.449 | 0.307 | 0.466 | 0.315 | 0.632 | 0.396 | 0.626 | 0.382 | 0.660 | 0.408 | 0.864 | 0.472 |
| Weather | 96 | 0.158 | 0.211 | **0.149** | **0.198** | 0.172 | 0.220 | 0.166 | 0.221 | 0.176 | 0.237 | 0.197 | 0.281 | 0.217 | 0.296 | 0.266 | 0.336 | 0.300 | 0.384 |
| | 192 | 0.204 | 0.251 | **0.194** | **0.241** | 0.219 | 0.261 | 0.207 | 0.261 | 0.220 | 0.282 | 0.237 | 0.312 | 0.276 | 0.336 | 0.307 | 0.367 | 0.598 | 0.544 |
| | 336 | 0.249 | 0.290 | **0.245** | **0.282** | 0.280 | 0.306 | 0.251 | 0.298 | 0.265 | 0.319 | 0.298 | 0.353 | 0.339 | 0.380 | 0.359 | 0.395 | 0.578 | 0.523 |
| | 720 | **0.313** | 0.335 | 0.314 | **0.334** | 0.365 | 0.359 | 0.301 | 0.338 | 0.323 | 0.362 | 0.352 | 0.388 | 0.403 | 0.428 | 0.419 | 0.428 | 1.059 | 0.741 |

† For PatchTST, we choose the model showing the best MSE across PatchTST/64 and PatchTST/42.

Table 1: Multivariate long-term forecasting MSE and MAE. Forecasting horizon $H \in \{24, 36, 48, 60\}$ for ILI dataset and $H \in \{96, 192, 336, 720\}$ for the other datasets. The look-back window size $L$ is set differently for each model, following the original setting of each model. The **best results** and the second best results are highlighted in **bold** and underlined, respectively.

While the look-back window serves as a crucial factor in providing information about the time series, the performances of the models has often been compared without accounting for variations in input length. For example, In Table 1, Informer, Autoformer, and FEDformer employ an input length of 96, DLinear utilizes an input length of 336, and PatchTST/64 employs an input length of 512 in all

| Method | | LGPred | | PatchTST† | | DeepTime | | DLinear | | ETSformer | | FEDformer | | Autoformer | | Informer | |
|---|---|---|---|---|---|---|---|---|---|---|---|---|---|---|---|---|---|
| Metric | | MSE | MAE | MSE | MAE | MSE | MAE | MSE | MAE | MSE | MAE | MSE | MAE | MSE | MAE | MSE | MAE |
| ETTh1 | 96 | **0.052** | **0.175** | 0.055 | 0.179 | - | - | 0.056 | 0.180 | - | - | 0.079 | 0.215 | 0.071 | 0.206 | 0.193 | 0.377 |
| | 192 | **0.065** | **0.196** | 0.071 | 0.205 | - | - | 0.071 | 0.204 | - | - | 0.104 | 0.245 | 0.114 | 0.262 | 0.217 | 0.395 |
| | 336 | **0.076** | **0.219** | **0.076** | 0.220 | - | - | 0.098 | 0.244 | - | - | 0.119 | 0.270 | 0.107 | 0.258 | 0.202 | 0.381 |
| | 720 | **0.079** | **0.224** | 0.087 | 0.232 | - | - | 0.189 | 0.359 | - | - | 0.142 | 0.299 | 0.126 | 0.283 | 0.183 | 0.355 |
| ETTh2 | 96 | **0.125** | **0.268** | 0.129 | 0.282 | - | - | 0.131 | 0.279 | - | - | 0.128 | 0.271 | 0.153 | 0.306 | 0.213 | 0.373 |
| | 192 | **0.162** | **0.320** | 0.168 | 0.328 | - | - | 0.176 | 0.329 | - | - | 0.185 | 0.330 | 0.204 | 0.351 | 0.227 | 0.387 |
| | 336 | 0.173 | 0.337 | **0.171** | **0.336** | - | - | 0.209 | 0.367 | - | - | 0.231 | 0.378 | 0.246 | 0.389 | 0.242 | 0.401 |
| | 720 | **0.217** | **0.376** | 0.223 | 0.380 | - | - | 0.276 | 0.420 | - | - | 0.278 | 0.420 | 0.268 | 0.409 | 0.291 | 0.439 |
| ETTm1 | 96 | **0.026** | **0.121** | **0.026** | **0.121** | - | - | 0.028 | 0.123 | - | - | 0.033 | 0.140 | 0.056 | 0.183 | 0.109 | 0.277 |
| | 192 | **0.038** | **0.149** | 0.039 | 0.150 | - | - | 0.045 | 0.156 | - | - | 0.058 | 0.186 | 0.081 | 0.216 | 0.151 | 0.310 |
| | 336 | **0.051** | **0.172** | 0.053 | 0.173 | - | - | 0.061 | 0.182 | - | - | 0.084 | 0.231 | 0.076 | 0.218 | 0.427 | 0.591 |
| | 720 | **0.067** | **0.196** | 0.073 | 0.206 | - | - | 0.080 | 0.210 | - | - | 0.102 | 0.250 | 0.110 | 0.267 | 0.438 | 0.586 |
| ETTm2 | 96 | **0.062** | **0.183** | 0.065 | 0.186 | 0.065 | 0.186 | 0.063 | **0.183** | 0.080 | 0.212 | 0.067 | 0.198 | 0.065 | 0.189 | 0.088 | 0.225 |
| | 192 | **0.088** | **0.227** | 0.093 | 0.231 | 0.096 | 0.234 | 0.092 | **0.227** | 0.150 | 0.302 | 0.102 | 0.245 | 0.118 | 0.256 | 0.132 | 0.283 |
| | 336 | **0.114** | **0.260** | 0.120 | 0.265 | 0.138 | 0.285 | 0.119 | 0.261 | 0.175 | 0.334 | 0.130 | 0.279 | 0.154 | 0.305 | 0.180 | 0.336 |
| | 720 | **0.144** | **0.303** | 0.171 | 0.322 | 0.186 | 0.338 | 0.175 | 0.320 | 0.224 | 0.379 | 0.178 | 0.325 | 0.182 | 0.335 | 0.300 | 0.435 |

† For PatchTST, we choose the model showing the best MSE across PatchTST/64 and PatcthTST/42.

Table 2: Univariate long-term forecasting MSE and MAE. Forecasting horizon $H \in \{96, 192, 336, 720\}$ for all ETT variants. The look-back window size $L$ is set differently for each model, following the original setting of each model. The **best results** and the second best results are highlighted in **bold** and underlined, respectively.

datasets except ILI. In case of DeepTime, ETSformer and our LGPred in Table 1, the input length is treated as a hyperparameter. Table 3 presents the long-term forecasting performance assessed with a fixed input length (36 for ILI, 96 for the others), averaged over four distinct forecasting horizons, consistent with a recent paper of (Wu et al., 2022). In this experimental setup, our LGPred achieves the best performances in 7 and 6 out of 9 datasets for MSE and MAE metric each, reaffirming the effectiveness of LGPred even in scenarios where the input length is limited.

| Method | LGPred | | PatchTST | | TimesNet | | ETSformer | | DLinear | | FEDformer | | NSformer | | Autoformer | | Informer | |
|---|---|---|---|---|---|---|---|---|---|---|---|---|---|---|---|---|---|---|
| Metric | MSE | MAE | MSE | MAE | MSE | MAE | MSE | MAE | MSE | MAE | MSE | MAE | MSE | MAE | MSE | MAE | MSE | MAE |
| Electricity | **0.188** | **0.279** | 0.204 | 0.291 | 0.192 | 0.295 | 0.208 | 0.323 | 0.212 | 0.300 | 0.214 | 0.327 | 0.193 | 0.296 | 0.227 | 0.338 | 0.311 | 0.397 |
| ETTh1 | **0.438** | **0.429** | 0.447 | 0.442 | 0.458 | 0.450 | 0.542 | 0.510 | 0.456 | 0.452 | 0.440 | 0.460 | 0.570 | 0.537 | 0.496 | 0.487 | 1.040 | 0.795 |
| ETTh2 | **0.370** | **0.395** | 0.377 | 0.403 | 0.414 | 0.427 | 0.439 | 0.452 | 0.559 | 0.515 | 0.437 | 0.449 | 0.526 | 0.516 | 0.450 | 0.459 | 4.431 | 1.729 |
| ETTm1 | 0.398 | **0.402** | **0.391** | **0.402** | 0.400 | 0.406 | 0.429 | 0.425 | 0.403 | 0.407 | 0.448 | 0.452 | 0.481 | 0.456 | 0.588 | 0.517 | 0.961 | 0.734 |
| ETTm2 | **0.280** | 0.329 | 0.283 | **0.327** | 0.291 | 0.333 | 0.293 | 0.342 | 0.350 | 0.401 | 0.305 | 0.349 | 0.306 | 0.347 | 0.327 | 0.371 | 1.410 | 0.810 |
| Exchange | **0.223** | **0.338** | 0.372 | 0.407 | 0.416 | 0.443 | 0.410 | 0.427 | 0.354 | 0.414 | 0.519 | 0.500 | 0.461 | 0.454 | 0.613 | 0.539 | 1.550 | 0.998 |
| ILI | **1.974** | **0.905** | 2.206 | 0.913 | 2.139 | 0.931 | 2.497 | 1.004 | 2.616 | 1.090 | 2.847 | 1.144 | 2.077 | 0.914 | 3.006 | 1.161 | 5.137 | 1.544 |
| Traffic | **0.469** | 0.325 | 0.481 | **0.304** | 0.620 | 0.336 | 0.621 | 0.396 | 0.625 | 0.383 | 0.610 | 0.376 | 0.624 | 0.340 | 0.628 | 0.379 | 0.764 | 0.416 |
| Weather | 0.259 | 0.290 | **0.258** | **0.280** | 0.259 | 0.287 | 0.271 | 0.334 | 0.265 | 0.317 | 0.309 | 0.360 | 0.288 | 0.314 | 0.338 | 0.382 | 0.634 | 0.548 |

Table 3: Multivariate long-term forecasting comparison with the fixed look-back window size. MSE and MAE scores are averaged from 4 different forecasting horizons. Forecasting horizon $H \in \{24, 36, 48, 60\}$ for ILI dataset and $H \in \{96, 192, 336, 720\}$ for the other datasets. Look-back window size $L$ is fixed to 36 for ILI dataset and 96 for the other datasets. The **best results** and the second best results are highlighted in **bold** and underlined, respectively.

## 5.3 Experiment with Synthetic Data

We mention that the LGPred enables the dynamic adaptation for the linear predictor, which is an inherent limitation of the linear predictor. To verify our claim, we conduct an additional experiment with synthetic data generated with TimeSynth (Maat et al., 2017). We randomly generate the multiple synthetic time series data with diverse trends and seasonalities, and train the predictor across the various synthetic time series. For trend, we generate a sinusoidal time series with a fixed amplitude of 1.0, and a frequency sampled uniformly from low frequency range of $[10^{-5}, 10^{-4}]$. The seasonality is generated as a combination of two sinusoidal time series with varying amplitudes sampled from range of $[0.02, 0.1]$ and frequencies sampled from the high frequency range of $[0.01, 1]$. For experiment, we generate 20 different time series with varying trends and seasonalities. Each time series comprises 10,000 timestamps, partitioned into training, validation, and test sets in a 7:1:2 ratio. Since no noise is added to our synthetic data, the linear predictor can easily forecast individual time series. However, when presented with multiple time series exhibiting distinct characteristics, the linear predictor fails to learn the characteristic of the multiple time series. In contrast, our LGPred is able to adapt to diverse characterics of the time series, enabling accurate predictions across multiple time series. Experimental results in Table 4 clearly show that Linear models fails in forecasting synthetic data with varying characteristics, while LGPred successfully forecasts the synthetic data showing extremely low errors.

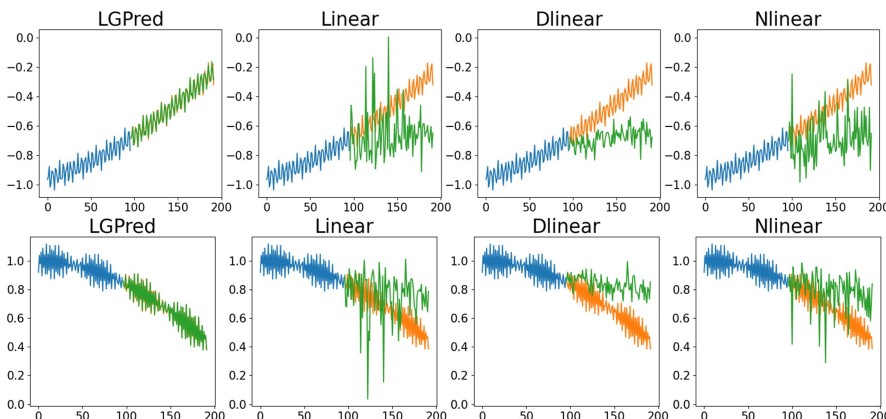

Figure 3: Visualization of prediction results from synthetic data experiments. The blue line indicates the input look-back window, the orange line is the groundtruth of the forecasting horizon, and the green line is the prediction result of each model.

In Figure 3, we visualize the forecasting result of Linear models and LGPred. Despite clear trends and seasonalities in the input data, Linear models fail to predict the future values. LGPred, on the other hand, provides precise future predictions, accurately identifying trends and seasonalities. From this experiment, we can clearly see that our LGPred framework effectively addresses the limitation of the Linear model, enhancing its adaptability to dynamic time series.

| Method | LGPred | | Linear | | DLinear | | NLinear | |
|--------|--------|--------|--------|--------|--------|--------|--------|--------|
| Metric | MSE | MAE | MSE | MAE | MSE | MAE | MSE | MAE |
| 96 | **0.00047** | **0.016** | 0.097 | 0.248 | 0.087 | 0.233 | 0.019 | 0.108 |
| 192 | **0.0019** | **0.031** | 0.125 | 0.284 | 0.122 | 0.280 | 0.055 | 0.167 |
| 336 | **0.0056** | **0.053** | 0.161 | 0.331 | 0.157 | 0.326 | 0.111 | 0.237 |
| 720 | **0.030** | **0.118** | 0.225 | 0.379 | 0.220 | 0.375 | 0.298 | 0.401 |

Table 4: Experiment results from synthetic data. Forecasting Horizon $H \in \{96, 192, 336, 720\}$. Look-back window size $L$ is equal to $H$. The best results are highlighted in **bold**.

## 6 DISCUSSION

This paper proposes the LGPred framework as an innovative approach to generate network parameters for time series forecasting. While our experiments demonstrate state-of-the-art performance, our approach has some limitations. In this paper, we focus solely on linear predictors, which lack the capability for multivariate forecasting, predicting different channels independently instead of utilizing all available information. As a result, LGPred does not fully utilize the potential of multivariate information, leaving room for further improvement in forecasting performance. We believe that to overcome this limitation, it is necessary to explore the potential of our methodology by integrating it with predictor architectures that support multivariate forecasting capabilities, and future works can be made to leverage the LGPred framework in various architectures for better forecasting results.

## 7 CONCLUSION

In this paper, we introduce LGPred, a novel framework for time series forecasting that can generate a predictor adaptively to the given time series. Proposed LGPred incorporates time series decomposition and representation modules tailored to each component to capture the characteristics of time series more effectively. We adopt a template predictor with a bottleneck architecture to efficiently generate the predictor as well as leveraging the shared information across time series. Our extensive experiments demonstrate that LGPred outperforms existing state-of-the-art forecasting methods on six benchmark datasets. Moreover, we believe that our predictor generation method can be applied to various forecasting architectures and can lead to valuable future works.

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

## A    EXPERIMENTAL DETAILS

### A.1    DATASET DESCRIPTION

In Table 5, we present a more comprehensive overview of the ECL [1], ETT [2], Exchange[3], ILI[4], Traffic[5], and Weather[6] datasets utilized in our experiments. The "Length" column denotes the total number of timestamps in each dataset, while the "Dimension" column indicates the number of features present in the dataset. Additionally, the "Unit" column represents the time interval at which each timestamp is collected. For the experiment, we divide each dataset into training, validation, and test sets by the ratio of 7:1:2.

| Dataset | Length | Dimension | Unit |
|---------|--------|-----------|------|
| ECL | 26304 | 321 | 1 Hour |
| ETTh1 | 17420 | 7 | 1 Hour |
| ETTh2 | 17420 | 7 | 1 Hour |
| ETTm1 | 69680 | 7 | 15 Minutes |
| ETTm2 | 69680 | 7 | 15 Minutes |
| Exchange | 7588 | 8 | 1 Day |
| ILI | 966 | 7 | 1 Week |
| Traffic | 17544 | 862 | 1 Hour |
| Weather | 52696 | 21 | 10 Minutes |

Table 5: Dataset Description

### A.2    HYPERPARAMETER DETAILS

Here, we present the hyperparameter settings of the proposed LGPred for the experiments discussed in Section 5 of the main paper. We consider following 8 hyperparameters; the input sequence length $L$, latent dimension size $d_{latent}$ for $W_{gen}$, feature size $d_{feat}$ for predictor generation, representation dimension size $d_{feat}$, kernel size of convolution layers $k$ in seasonality representation module, number of layers $N_L$ in trend and seasonality representation modules, learning rate $lr$ and dropout ratio $dr$.

It is importnat to note that the same number of layers is used for both trend and seasonality representation modules. For regularization purpose, we apply dropout with a ratio $dr$ on the generated weight $W_{gen}$ and bias $b_{gen}$. In Table 6, we report the search range for each hyperparameter. Hyperparameter tuning is performed using a grid search methodology, where we systematically explore the reported search ranges for each hyperparameter. By evaluating the performance of the model using the Mean Squared Error (MSE) metric, we identify the hyperparameter set that achieves the best MSE score for each experiment. This process allows us to optimize the model's performance and select the most suitable hyperparameters for our experiments.

## B    ADDITIONAL EXPERIMENT RESULTS

### B.1    ABLATION STUDY

In this section, we conduct an ablation study to investigate the individual contributions and significance of the component modules within our LGPred framework. The study involves examining the effects of the decomposition module, representation module, and template predictor. First, to ablate decomposition module, we evaluate the variant of LGPred generating the predictor using the representation obtained from the original input time series. This variant utilizes the original time

---

[1]https://archive.ics.uci.edu/ml/datasets/ElectricityLoadDiagrams20112014

[2]https://github.com/zhouhaoyi/ETDataset

[3]https://github.com/laiguokun/multivariate-time-series-data

[4]https://gis.cdc.gov/grasp/fluview/fluportaldashboard.html

[5]https://pems.dot.ca.gov/

[6]https://www.bgc-jena.mpg.de/wetter/

| Hyperparameter | Range | |
|---|---|---|
| | General | ILI |
| Sequence Length $L$ | $\{96, 192, 336, 720\}$ | $\{24, 36, 48, 60, 104\}$ |
| Latent Dimension $d_{latent}$ | $\{16, 32, 64, 128, 256\}$ | $\{8, 16, 32, 64\}$ |
| Feature Dimension $d_{feat}$ | $\{32, 64, 128, 256, 512, 1024\}$ | |
| Representation Dimension $d_{rep}$ | $\{4, 8, 16, 32, 64, 128, 256, 512\}$ | |
| Convolution kernel size $k$ | $\{3, 5, 7, 9, 11\}$ | |
| Number of Layers $N_L$ | $\{1, 2, 3, 4, 5\}$ | |
| Learning Rate $lr$ | $\{10^{-2}, 10^{-3}, 3 \times 10^{-4}, 10^{-4}, 3 \times 10^{-5}, 10^{-5}\}$ | |
| Dropout Ratio $dr$ | $\{0.0, 0.1, 0.2, 0.3, 0.4\}$ | |

Table 6: Ranges of hyperparameters in LGPred

series as input for both the trend and seasonality representation modules. Second, to investigate the contribution of the representation module, we isolate the representation module and directly generate predictors from the decomposed trend and seasonality components using the predictor generator. Lastly, we analyze the influence of the template predictor by generating a full predictor using the predictor generator, thereby evaluating its effects. Table 7 displays the results of ablation experiments, with the DLinear baseline results. From the experiment results, one can observe that all the components play an important roles in forecasting performance. Notably, the removal of the representation modules results in a significant degradation of performance, highlighting the importance of obtaining proper representations for effectively capturing the characteristics of a time series. Regarding the impact of the decomposition module and template predictor, our ablation experiments demonstrate varying effects across different datasets. While the decomposition module exhibits a more pronounced influence in enhancing performance compared to the baseline for the Exchange and Traffic datasets, the template predictor proves to be more significant in the ILI and Weather datasets.

| Dataset | Electricity | | Exchange | | ILI | | Traffic | | Weather | |
|---|---|---|---|---|---|---|---|---|---|---|
| Metric | MSE | MAE | MSE | MAE | MSE | MAE | MSE | MAE | MSE | MAE |
| DLinear | 0.203 | 0.301 | 0.643 | 0.6011 | 2.368 | 1.096 | 0.466 | 0.315 | 0.323 | 0.362 |
| LGPred w/o D | 0.201 | 0.296 | 1.132 | 0.796 | 1.790 | 0.926 | 0.630 | 0.442 | 0.323 | 0.343 |
| LGPred w/o R | 3.065 | 1.329 | 1.729 | 1.701 | 2.408 | 1.104 | 6.277 | 1.304 | 0.507 | 0.447 |
| LGPred w/o T | 0.201 | 0.300 | 0.540 | 0.564 | 3.738 | 1.306 | 0.465 | 0.324 | 0.334 | 0.352 |
| LGPred | **0.189** | **0.290** | **0.387** | **0.493** | **1.549** | **0.853** | **0.430** | **0.301** | **0.313** | **0.335** |

Table 7: Ablation studies for component modules of LGPred on multivariate long-term forecasting. Forecasting horizon $H = 60$ for ILI dataset and $H = 720$ for the other datasets. D, R, and T denotes decomposition module, representation module, and template predictor. The **best results** are highlighted in **bold**.

In order to examine the impact of each decomposed component time series in LGPred, we further conduct another ablation experiment. In Table 8, the performance of LGPred generating predictors using either only the trend or only the seasonality component, is evaluated and presented. Our findings reveal that the contributions of the different components primarily depend on the characteristics of the dataset. For instance, in the ECL dataset where a clear trend is absent and seasonal patterns are prominent as illustrated in Figure 4, LGPred utilizing only the seasonality component outperforms LGPred utilizing the trend component. Conversely, in the Exchange dataset characterized by a distinct tendency and low periodicity as depicted in Figure 4, LGPred employing only the trend component exhibits better performance than LGPred using seasonality component.

| Dataset | Electricity | | Exchange | | ILI | | Traffic | | Weather | |
|---|---|---|---|---|---|---|---|---|---|---|
| Metric | MSE | MAE | MSE | MAE | MSE | MAE | MSE | MAE | MSE | MAE |
| LGPred Trend only | 0.209 | 0.302 | 0.425 | 0.506 | 2.189 | 1.058 | 0.444 | 0.317 | 0.320 | 0.340 |
| LGPred Seasonality only | 0.191 | 0.290 | 0.999 | 0.757 | 1.897 | 0.964 | 0.470 | 0.318 | 0.319 | 0.341 |
| LGPred | **0.189** | **0.290** | **0.387** | **0.493** | **1.549** | **0.853** | **0.430** | **0.301** | **0.313** | **0.335** |

Table 8: Ablation studies for decomposed component time series on multivariate long-term forecasting. Forecasting horizon $H = 60$ for ILI dataset and $H = 720$ for the other datasets. The **best results** are highlighted in **bold**.

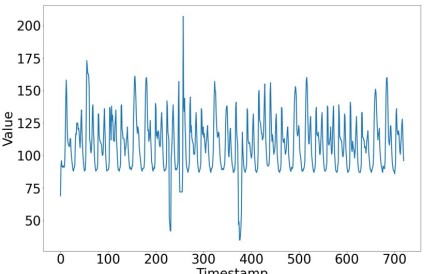
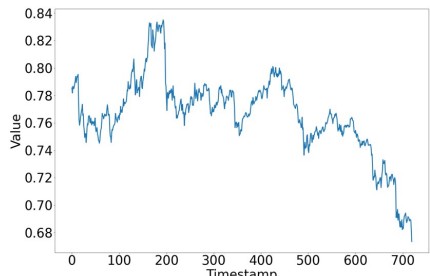

Figure 4: Visualization of ECL (left) and Exchange (right) datasets

## B.2 Applying LGPred to Existing Forecasting Models

In Section 3 of main paper, we introduce the LGPred framework, which dynamically generates a linear predictor tailored to the input data. While the generation of the linear predictor alone achieves state-of-the-art forecasting performance, the efficacy of the proposed LGPred framweork is not limited to the linear predictors only. Given that the fully-connected layer serves as a fundamental component widely employed in numerous forecasting models, our LGPred framework can be easily integrated with existing forecasting models to enhance their forecasting capabilities. To demonstrate the applicability of LGPred, we incorporate it into two state-of-the-art forecasting models, PatchTST (Nie et al., 2022) and TimesNet (Wu et al., 2022). PatchTST utilizes a transformer backbone for feature extraction and employs a linear head for final forecasting. We apply LGPred into PatchTST by generating the final linear head using LGPred. On the other hand, TimesNet conducts prediction from the input window to the output horizon at the first fully-connected predictor, followed by a refinement step using TimesBlocks designed with Inception blocks. In the case of TimesNet, we apply our LGPred by generating the first predictor. The performance comparison between the original forecasting models and the combinations with LGPred is presented in Table 9, utilizing official implementation codes, original scripts and hyperparameters provided by the authors [7][8]. For PatchTST, the combination with LGPred yields performance levels similar to the base model. However, in experiments with a long horizon ($H = 720$), the combined model clearly outperforms the base PatchTST. On the other hand, when integrated with TimesNet, LGPred consistently enhances the performance of the base model. Particularly in the Exchange dataset, the combination with LGPred reduces the Mean Squared Error (MSE) by approximately 20% in $H = 192$ ($0.240 \rightarrow 0.185$, -22.92%), $H = 336$ ($0.362 \rightarrow 0.296$, -18.23% ), and $H = 720$ ($0.930 \rightarrow 0.741$, -20.32%) cases.

| Method | | PatchTST | | LGPred+PatchTST | | TimesNet | | LGPred+TimesNet | |
|---|---|---|---|---|---|---|---|---|---|
| Metric | | MSE | MAE | MSE | MAE | MSE | MAE | MSE | MAE |
| Electricity | 96 | 0.130 | 0.222 | 0.129 | 0.228 | 0.168 | 0.272 | 0.163 | 0.267 |
| | 192 | 0.148 | 0.240 | 0.145 | 0.244 | 0.189 | 0.291 | 0.172 | 0.275 |
| | 336 | 0.165 | 0.259 | 0.160 | 0.261 | 0.205 | 0.306 | 0.186 | 0.290 |
| | 720 | 0.210 | 0.298 | 0.186 | 0.283 | 0.214 | 0.313 | 0.213 | 0.311 |
| Exchange | 96 | - | - | - | - | 0.105 | 0.235 | 0.093 | 0.218 |
| | 192 | - | - | - | - | 0.240 | 0.356 | 0.185 | 0.312 |
| | 336 | - | - | - | - | 0.362 | 0.438 | 0.296 | 0.394 |
| | 720 | - | - | - | - | 0.930 | 0.734 | 0.741 | 0.660 |
| Traffic | 96 | 0.365 | 0.250 | 0.369 | 0.268 | 0.589 | 0.315 | 0.589 | 0.320 |
| | 192 | 0.383 | 0.258 | 0.385 | 0.270 | 0.616 | 0.323 | 0.607 | 0.325 |
| | 336 | 0.396 | 0.264 | 0.397 | 0.286 | 0.634 | 0.340 | 0.630 | 0.333 |
| | 720 | 0.435 | 0.287 | 0.428 | 0.306 | 0.660 | 0.350 | 0.652 | 0.350 |
| Weather | 96 | 0.150 | 0.198 | 0.147 | 0.196 | 0.170 | 0.220 | 0.163 | 0.218 |
| | 192 | 0.196 | 0.242 | 0.193 | 0.241 | 0.226 | 0.265 | 0.214 | 0.259 |
| | 336 | 0.250 | 0.284 | 0.243 | 0.286 | 0.281 | 0.304 | 0.268 | 0.300 |
| | 720 | 0.318 | 0.335 | 0.312 | 0.333 | 0.359 | 0.354 | 0.340 | 0.348 |

Table 9: Multivariate long-term forecasting MSE and MAE comparison between base forecasting models and LGPred-integrated forecasting models. Forecasting horizon $H \in \{96, 192, 336, 720\}$
.

---

[7]https://github.com/yuqinie98/PatchTST

[8]https://github.com/thuml/TimesNet

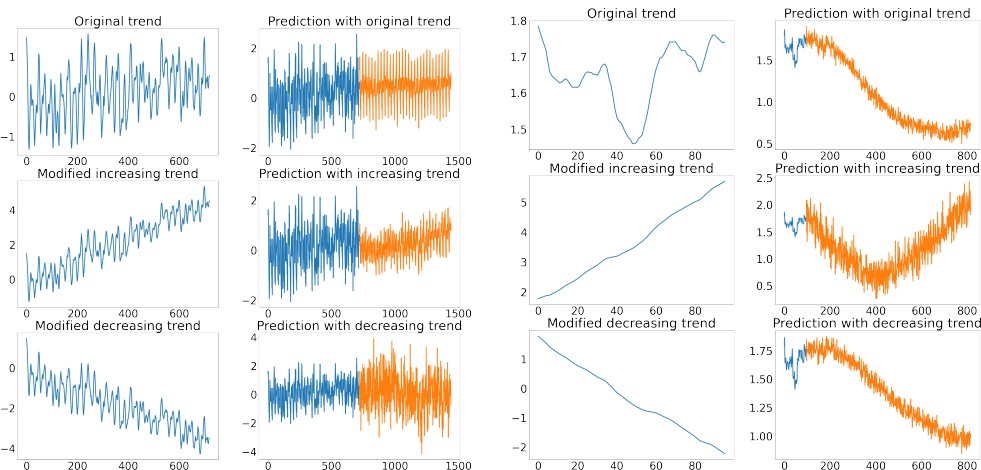

Figure 5: Visualization of prediction results using modified trends with increasing (2nd row) and decreasing (3rd row) tendency on ECL (left) and Exchange (right) datasets

## B.3 EXPERIMENT WITH MODIFIED TREND

In the main paper, we argue that the proposed LGPred framework can reflect the characteristics of the dynamically changing time series to prediction. To validate our claim, we conduct experiments to assess the capabilities of LGPred by introducing modified trends or seasonality to the representation modules and observing the resulting impact on predictions. Among trend and seasonality, we choose to focus on trend as it allows for a clearer demonstration of how modifications affect predictions. The results of our experiments on the ECL and Exchange datasets, involving two distinct types of modified trends (one increasing and the other decreasing), are presented in Figure 5. In the case of ECL dataset where the LGPred with original trend yields a relatively flat prediction, we observe that introducing an increasing trend leads to an upward prediction, while injecting a decreasing trend results in a downward prediction. For Exchange dataset, the prediction with original trend descends at first and starts to rise at the end. In this case, incorporating an increasing trend induces an earlier ascent, whereas employing a decreasing trend prolongs the monotonous descent until the end. These results affirm that the proposed LGPred is able to effectively capture the characteristics of the time series and reflect them in the prediction process.

## B.4 FULL RESULTS OF TABLE 3

In Table 10, we display the full results of multivariate long-term forecasting experiments with fixed look-back window size. We can see that our LGPred achieves the best performances in 21 and 20 cases out of 36 experiments for MSE and MAE metric each, reaffirming that our LGPred performs better than the baselines such as PatchTST and TimesNet.

## B.5 RANDOM CONTROL EXPERIMENTS

In the main paper, we report the experiment results measured with the fixed random seed. To verify the robustness of our LGPred against randomness, we conduct additional experiments with 3 different seeds. Table 11, shows the experiment results on multivaraite long-term forecasting. We display the mean and the standard deviation of MSE and MAE scores, calculated from 3 experiments with different random seeds. The results confirm that proposed LGPred shows good performances regardless of initialization.

## B.6 HYPERPARAMETER SENSITIVITY ANALYSIS

In this section, we analyze proposed LGPred to figure out which hyperparameter affects the performance of LGPred. We conducts an experiments with 4 hyperparameters related to prediction and predictor generation; sequence length $L$, latent dimension $d_{latent}$, representation dimension $d_{rep}$,

Table 10 — Multivariate long-term forecasting MSE and MAE.

| Method | | LGPred | | PatchTST | | TimesNet | | ETSformer | | DLinear | | FEDformer | | NSformer | | Autoformer | | Informer | |
|---|---|---|---|---|---|---|---|---|---|---|---|---|---|---|---|---|---|---|---|
| Metric | | MSE | MAE | MSE | MAE | MSE | MAE | MSE | MAE | MSE | MAE | MSE | MAE | MSE | MAE | MSE | MAE | MSE | MAE |
| Electricity | 96 | **0.167** | **0.257** | 0.181 | 0.271 | 0.168 | 0.272 | 0.187 | 0.304 | 0.197 | 0.282 | 0.193 | 0.308 | 0.169 | 0.273 | 0.201 | 0.317 | 0.274 | 0.368 |
| Electricity | 192 | **0.174** | **0.267** | 0.187 | 0.276 | 0.184 | 0.289 | 0.199 | 0.315 | 0.196 | 0.285 | 0.201 | 0.315 | 0.182 | 0.286 | 0.222 | 0.334 | 0.296 | 0.386 |
| Electricity | 336 | **0.185** | **0.280** | 0.204 | 0.292 | 0.198 | 0.300 | 0.212 | 0.329 | 0.209 | 0.301 | 0.214 | 0.329 | 0.200 | 0.304 | 0.231 | 0.338 | 0.300 | 0.394 |
| Electricity | 720 | 0.224 | **0.313** | 0.245 | 0.325 | **0.220** | 0.320 | 0.233 | 0.345 | 0.245 | 0.333 | 0.246 | 0.355 | 0.222 | 0.321 | 0.254 | 0.361 | 0.373 | 0.439 |
| Electricity | Avg | **0.188** | **0.279** | 0.204 | 0.291 | 0.192 | 0.295 | 0.208 | 0.323 | 0.212 | 0.300 | 0.212 | 0.327 | 0.193 | 0.296 | 0.227 | 0.338 | 0.311 | 0.397 |
| ETTh1 | 96 | 0.381 | **0.393** | **0.376** | 0.395 | 0.384 | 0.402 | 0.494 | 0.479 | 0.386 | 0.400 | **0.376** | 0.419 | 0.513 | 0.491 | 0.449 | 0.459 | 0.865 | 0.713 |
| ETTh1 | 192 | 0.431 | **0.421** | 0.425 | 0.427 | 0.436 | 0.429 | 0.538 | 0.504 | 0.437 | 0.432 | **0.420** | 0.448 | 0.534 | 0.504 | 0.500 | 0.482 | 1.008 | 0.792 |
| ETTh1 | 336 | 0.470 | **0.441** | 0.460 | 0.447 | 0.491 | 0.469 | 0.574 | 0.521 | 0.481 | 0.459 | **0.459** | 0.465 | 0.588 | 0.535 | 0.521 | 0.496 | 1.107 | 0.809 |
| ETTh1 | 720 | **0.469** | **0.461** | 0.527 | 0.498 | 0.521 | 0.500 | 0.562 | 0.535 | 0.519 | 0.516 | 0.506 | 0.507 | 0.643 | 0.616 | 0.514 | 0.512 | 1.181 | 0.865 |
| ETTh1 | Avg | **0.438** | **0.429** | 0.447 | 0.442 | 0.458 | 0.450 | 0.542 | 0.510 | 0.456 | 0.452 | 0.440 | 0.460 | 0.570 | 0.537 | 0.496 | 0.487 | 1.040 | 0.795 |
| ETTh2 | 96 | **0.289** | **0.336** | 0.291 | 0.342 | 0.340 | 0.374 | 0.340 | 0.391 | 0.333 | 0.387 | 0.358 | 0.397 | 0.476 | 0.458 | 0.346 | 0.388 | 3.755 | 1.525 |
| ETTh2 | 192 | **0.366** | **0.384** | 0.377 | 0.398 | 0.402 | 0.414 | 0.430 | 0.439 | 0.477 | 0.476 | 0.429 | 0.439 | 0.512 | 0.493 | 0.456 | 0.452 | 5.602 | 1.931 |
| ETTh2 | 336 | **0.406** | **0.420** | 0.414 | 0.427 | 0.452 | 0.452 | 0.486 | 0.479 | 0.594 | 0.541 | 0.496 | 0.487 | 0.552 | 0.551 | 0.482 | 0.486 | 4.721 | 1.835 |
| ETTh2 | 720 | **0.418** | **0.438** | 0.424 | 0.443 | 0.462 | 0.468 | 0.500 | 0.497 | 0.831 | 0.657 | 0.463 | 0.474 | 0.562 | 0.560 | 0.515 | 0.511 | 3.647 | 1.625 |
| ETTh2 | Avg | **0.370** | **0.395** | 0.377 | 0.403 | 0.414 | 0.427 | 0.439 | 0.452 | 0.559 | 0.515 | 0.437 | 0.449 | 0.526 | 0.516 | 0.450 | 0.459 | 4.431 | 1.729 |
| ETTm1 | 96 | 0.345 | **0.369** | 0.339 | 0.372 | **0.338** | 0.375 | 0.375 | 0.398 | 0.345 | 0.372 | 0.379 | 0.419 | 0.386 | 0.398 | 0.505 | 0.475 | 0.672 | 0.571 |
| ETTm1 | 192 | **0.352** | **0.377** | 0.369 | 0.387 | 0.374 | 0.387 | 0.408 | 0.410 | 0.380 | 0.389 | 0.426 | 0.441 | 0.459 | 0.444 | 0.553 | 0.496 | 0.795 | 0.669 |
| ETTm1 | 336 | 0.413 | 0.412 | **0.397** | **0.405** | 0.410 | 0.411 | 0.435 | 0.428 | 0.413 | 0.413 | 0.445 | 0.459 | 0.495 | 0.464 | 0.621 | 0.537 | 1.212 | 0.871 |
| ETTm1 | 720 | 0.480 | 0.449 | **0.459** | **0.443** | 0.478 | 0.450 | 0.499 | 0.462 | 0.474 | 0.453 | 0.543 | 0.490 | 0.585 | 0.516 | 0.671 | 0.561 | 1.166 | 0.823 |
| ETTm1 | Avg | 0.398 | 0.402 | **0.391** | **0.402** | 0.400 | 0.406 | 0.429 | 0.425 | 0.403 | 0.407 | 0.448 | 0.452 | 0.481 | 0.456 | 0.588 | 0.517 | 0.961 | 0.734 |
| ETTm2 | 96 | 0.180 | 0.264 | **0.178** | **0.259** | 0.187 | 0.267 | 0.189 | 0.280 | 0.193 | 0.292 | 0.203 | 0.287 | 0.192 | 0.274 | 0.255 | 0.339 | 0.365 | 0.453 |
| ETTm2 | 192 | **0.242** | 0.303 | **0.242** | **0.301** | 0.249 | 0.309 | 0.253 | 0.319 | 0.284 | 0.362 | 0.269 | 0.328 | 0.280 | 0.339 | 0.281 | 0.340 | 0.533 | 0.563 |
| ETTm2 | 336 | **0.293** | **0.342** | 0.304 | 0.344 | 0.321 | 0.351 | 0.314 | 0.357 | 0.369 | 0.427 | 0.325 | 0.366 | 0.334 | 0.361 | 0.339 | 0.372 | 1.363 | 0.887 |
| ETTm2 | 720 | **0.404** | 0.406 | 0.407 | **0.402** | 0.408 | 0.403 | 0.414 | 0.413 | 0.554 | 0.522 | 0.421 | 0.415 | 0.417 | 0.413 | 0.433 | 0.432 | 3.379 | 1.338 |
| ETTm2 | Avg | **0.280** | 0.329 | 0.283 | **0.327** | 0.291 | 0.333 | 0.293 | 0.342 | 0.350 | 0.401 | 0.305 | 0.349 | 0.306 | 0.347 | 0.327 | 0.371 | 1.410 | 0.810 |
| Exchange | 96 | **0.078** | **0.195** | 0.087 | 0.205 | 0.107 | 0.234 | 0.085 | 0.204 | 0.088 | 0.218 | 0.148 | 0.278 | 0.111 | 0.237 | 0.197 | 0.323 | 0.847 | 0.752 |
| Exchange | 192 | **0.157** | **0.288** | 0.179 | 0.301 | 0.226 | 0.344 | 0.182 | 0.303 | 0.176 | 0.315 | 0.271 | 0.380 | 0.219 | 0.335 | 0.300 | 0.369 | 1.204 | 0.895 |
| Exchange | 336 | **0.230** | **0.355** | 0.316 | 0.407 | 0.367 | 0.448 | 0.348 | 0.428 | 0.313 | 0.427 | 0.460 | 0.500 | 0.421 | 0.476 | 0.509 | 0.524 | 1.672 | 1.036 |
| Exchange | 720 | **0.426** | **0.515** | 0.903 | 0.714 | 0.964 | 0.746 | 1.025 | 0.774 | 0.839 | 0.695 | 1.195 | 0.841 | 1.092 | 0.769 | 1.447 | 0.941 | 2.478 | 1.310 |
| Exchange | Avg | **0.223** | **0.338** | 0.372 | 0.407 | 0.416 | 0.443 | 0.410 | 0.427 | 0.354 | 0.414 | 0.519 | 0.500 | 0.461 | 0.454 | 0.613 | 0.539 | 1.550 | 0.998 |
| ILI | 96 | **1.916** | **0.867** | 2.202 | 0.889 | 2.317 | 0.934 | 2.527 | 1.020 | 2.400 | 1.040 | 3.228 | 1.260 | 2.294 | 0.945 | 3.483 | 1.287 | 5.764 | 1.677 |
| ILI | 192 | 1.878 | 0.880 | 2.330 | 0.930 | 1.972 | 0.920 | 2.615 | 1.007 | 2.646 | 1.088 | 2.679 | 1.080 | **1.825** | **0.848** | 3.103 | 1.148 | 4.755 | 1.467 |
| ILI | 336 | **1.952** | 0.908 | 2.258 | 0.924 | 2.238 | 0.940 | 2.359 | 0.972 | 2.614 | 1.086 | 2.622 | 1.078 | 2.010 | **0.900** | 2.669 | 1.085 | 4.763 | 1.469 |
| ILI | 720 | 2.151 | 0.966 | 2.035 | **0.908** | **2.027** | 0.928 | 2.487 | 1.016 | 2.804 | 1.146 | 2.857 | 1.157 | 2.178 | 0.963 | 2.770 | 1.125 | 5.264 | 1.564 |
| ILI | Avg | **1.974** | **0.905** | 2.206 | 0.913 | 2.139 | 0.931 | 2.497 | 1.004 | 2.616 | 1.090 | 2.847 | 1.144 | 2.077 | 0.914 | 3.006 | 1.161 | 5.137 | 1.544 |
| Traffic | 96 | **0.434** | 0.305 | 0.460 | **0.296** | 0.593 | 0.321 | 0.607 | 0.392 | 0.650 | 0.396 | 0.587 | 0.366 | 0.612 | 0.338 | 0.613 | 0.388 | 0.719 | 0.391 |
| Traffic | 192 | **0.443** | 0.305 | 0.466 | **0.296** | 0.617 | 0.336 | 0.621 | 0.399 | 0.598 | 0.370 | 0.604 | 0.373 | 0.613 | 0.340 | 0.616 | 0.382 | 0.696 | 0.379 |
| Traffic | 336 | **0.475** | 0.330 | 0.481 | **0.304** | 0.629 | 0.336 | 0.622 | 0.396 | 0.605 | 0.373 | 0.621 | 0.383 | 0.618 | 0.328 | 0.622 | 0.337 | 0.777 | 0.420 |
| Traffic | 720 | 0.524 | 0.358 | **0.515** | **0.321** | 0.640 | 0.350 | 0.632 | 0.396 | 0.645 | 0.394 | 0.626 | 0.382 | 0.653 | 0.355 | 0.660 | 0.408 | 0.864 | 0.472 |
| Traffic | Avg | **0.469** | 0.325 | 0.481 | **0.304** | 0.620 | 0.336 | 0.621 | 0.396 | 0.625 | 0.383 | 0.610 | 0.376 | 0.624 | 0.340 | 0.628 | 0.379 | 0.764 | 0.416 |
| Weather | 96 | 0.178 | 0.225 | 0.178 | **0.219** | **0.172** | 0.220 | 0.197 | 0.281 | 0.196 | 0.255 | 0.217 | 0.296 | 0.173 | 0.223 | 0.266 | 0.336 | 0.300 | 0.384 |
| Weather | 192 | 0.227 | 0.269 | 0.221 | **0.256** | **0.219** | 0.261 | 0.237 | 0.312 | 0.237 | 0.296 | 0.276 | 0.336 | 0.245 | 0.285 | 0.307 | 0.367 | 0.598 | 0.544 |
| Weather | 336 | 0.280 | 0.309 | **0.279** | **0.297** | 0.280 | 0.306 | 0.298 | 0.353 | 0.298 | 0.335 | 0.339 | 0.380 | 0.321 | 0.338 | 0.359 | 0.395 | 0.578 | 0.523 |
| Weather | 720 | 0.349 | 0.358 | 0.356 | **0.349** | 0.365 | 0.359 | 0.352 | 0.288 | **0.345** | 0.381 | 0.403 | 0.428 | 0.414 | 0.410 | 0.419 | 0.428 | 1.059 | 0.741 |
| Weather | Avg | 0.259 | 0.290 | **0.258** | **0.280** | 0.259 | 0.287 | 0.271 | 0.334 | 0.265 | 0.317 | 0.309 | 0.360 | 0.288 | 0.314 | 0.338 | 0.382 | 0.634 | 0.548 |
| 1st count | | 41 | | 21 | | 5 | | 0 | | 1 | | 3 | | 3 | | 0 | | 0 | |

Table 10: Multivariate long-term forecasting MSE and MAE. Forecasting horizon $H \in \{24, 36, 48, 60\}$ for ILI dataset and $H \in \{96, 192, 336, 720\}$ for the other datasets. Look-back window size $L$ is fixed to 36 for ILI dataset and 96 for the other datasets. The **best results** and the second best results are highlighted in **bold** and underlined, respectively. Avg is averaged from all 4 forecasting horizons.

and feature dimension $d_{feat}$. In Figure. 6, we display experiment results on ECL dataset. In each figure, we measure the MSE by changing only one type of hyperparameter while fixing the rest hyperparameters. The result indicates that our LGPred is sensitive to latent dimension $d_{latent}$ and sequence length $L$, while it is relatively insensitive to representation dimension $d_{rep}$ and feature dimension $d_{feat}$.

## B.7 COMPLEXITY ANALYSIS

In this section, we conduct an analysis on the complexity of LGPred based on the number of trainable parameters. The number of model parameters undergoes significantly variation depending on the hyperparameters such as $d_{latent}$, $d_{feat}$, $d_{rep}$, $k$, and $N_L$. For instance, in the models used in the experiments of Table 1, the number of model parameters fluctuates considerably, ranging from as low

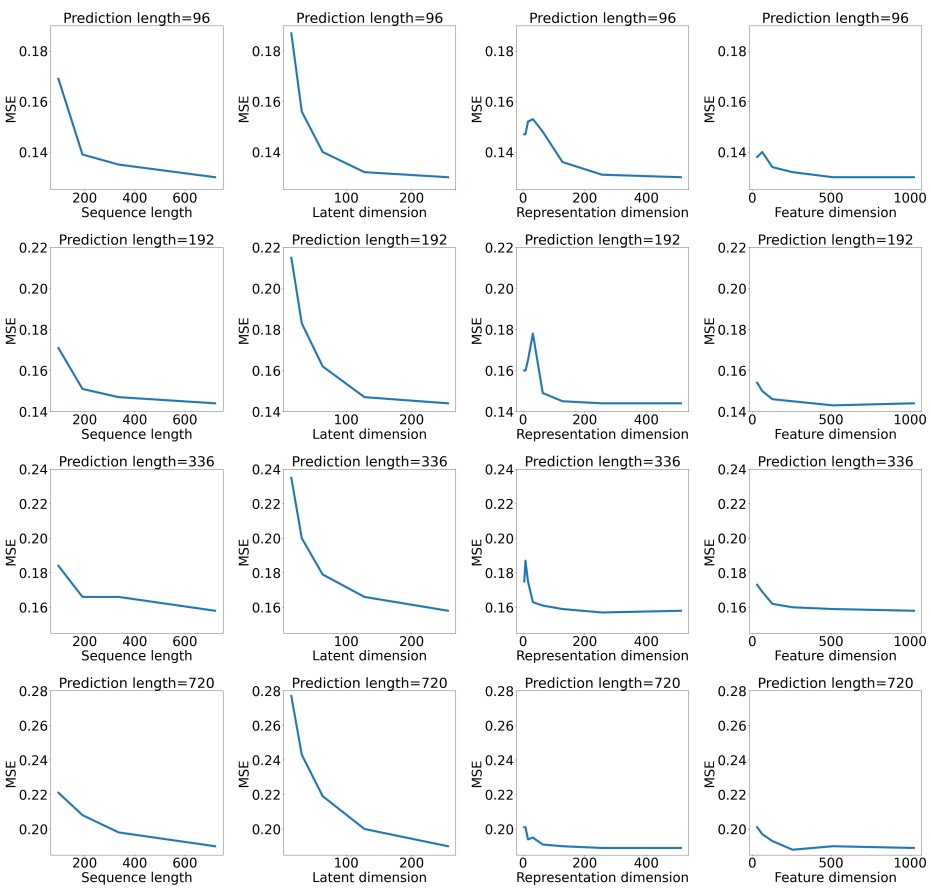

Figure 6: Visualization of hyperparameter sensitivity analysis results on ECL datasets.

| Method | LGPred | |
|---|---|---|
| Metric | MSE | MAE |
| Electricity 96 | 0.130 ± 0.001 | 0.230 ± 0.001 |
| Electricity 192 | 0.144 ± 0.002 | 0.245 ± 0.002 |
| Electricity 336 | 0.158 ± 0.001 | 0.260 ± 0.002 |
| Electricity 720 | 0.190 ± 0.001 | 0.290 ± 0.001 |
| ETTh1 96 | 0.375 ± 0.002 | 0.397 ± 0.002 |
| ETTh1 192 | 0.408 ± 0.001 | 0.417 ± 0.001 |
| ETTh1 336 | 0.437 ± 0.005 | 0.435 ± 0.005 |
| ETTh1 720 | 0.445 ± 0.005 | 0.461 ± 0.002 |
| ETTh2 96 | 0.273 ± 0.002 | 0.337 ± 0.001 |
| ETTh2 192 | 0.339 ± 0.004 | 0.380 ± 0.002 |
| ETTh2 336 | 0.362 ± 0.008 | 0.403 ± 0.003 |
| ETTh2 720 | 0.400 ± 0.013 | 0.442 ± 0.008 |
| ETTm1 96 | 0.305 ± 0.007 | 0.351 ± 0.003 |
| ETTm1 192 | 0.340 ± 0.007 | 0.369 ± 0.004 |
| ETTm1 336 | 0.384 ± 0.014 | 0.400 ± 0.007 |
| ETTm1 720 | 0.425 ± 0.003 | 0.423 ± 0.001 |
| ETTm2 96 | 0.165 ± 0.002 | 0.256 ± 0.003 |
| ETTm2 192 | 0.221 ± 0.003 | 0.295 ± 0.003 |
| ETTm2 336 | 0.275 ± 0.001 | 0.328 ± 0.001 |
| ETTm2 720 | 0.367 ± 0.007 | 0.392 ± 0.003 |
| Exchange 96 | 0.079 ± 0.001 | 0.195 ± 0.001 |
| Exchange 192 | 0.150 ± 0.008 | 0.288 ± 0.006 |
| Exchange 336 | 0.261 ± 0.002 | 0.378 ± 0.006 |
| Exchange 720 | 0.421 ± 0.004 | 0.516 ± 0.002 |
| ILI 96 | 1.707 ± 0.046 | 0.861 ± 0.020 |
| ILI 192 | 1.717 ± 0.013 | 0.845 ± 0.007 |
| ILI 336 | 1.727 ± 0.050 | 0.883 ± 0.029 |
| ILI 720 | 1.935 ± 0.058 | 0.941 ± 0.017 |
| Traffic 96 | 0.362 ± 0.006 | 0.271 ± 0.002 |
| Traffic 192 | 0.377 ± 0.002 | 0.281 ± 0.003 |
| Traffic 336 | 0.394 ± 0.003 | 0.294 ± 0.005 |
| Traffic 720 | 0.433 ± 0.004 | 0.308 ± 0.003 |
| Weather 96 | 0.162 ± 0.003 | 0.217 ± 0.001 |
| Weather 192 | 0.208 ± 0.001 | 0.256 ± 0.002 |
| Weather 336 | 0.253 ± 0.004 | 0.293 ± 0.005 |
| Weather 720 | 0.320 ± 0.003 | 0.343 ± 0.003 |

| Method | LGPred | |
|---|---|---|
| Metric | MSE | MAE |
| ETTh1 96 | 0.053 ± 0.001 | 0.177 ± 0.001 |
| ETTh1 192 | 0.068 ± 0.001 | 0.203 ± 0.002 |
| ETTh1 336 | 0.077 ± 0.001 | 0.221 ± 0.002 |
| ETTh1 720 | 0.079 ± 0.000 | 0.224 ± 0.000 |
| ETTh2 96 | 0.125 ± 0.000 | 0.268 ± 0.000 |
| ETTh2 192 | 0.165 ± 0.013 | 0.323 ± 0.012 |
| ETTh2 336 | 0.179 ± 0.007 | 0.343 ± 0.006 |
| ETTh2 720 | 0.222 ± 0.001 | 0.380 ± 0.001 |
| ETTm1 96 | 0.026 ± 0.000 | 0.121 ± 0.000 |
| ETTm1 192 | 0.039 ± 0.001 | 0.150 ± 0.001 |
| ETTm1 336 | 0.051 ± 0.001 | 0.172 ± 0.001 |
| ETTm1 720 | 0.072 ± 0.003 | 0.206 ± 0.004 |
| ETTm2 96 | 0.062 ± 0.001 | 0.185 ± 0.001 |
| ETTm2 192 | 0.090 ± 0.003 | 0.229 ± 0.003 |
| ETTm2 336 | 0.114 ± 0.003 | 0.263 ± 0.003 |
| ETTm2 720 | 0.155 ± 0.006 | 0.312 ± 0.006 |

Table 11: Multivariate (left) and Univariate (right) long-term forecasting MSE and MAE with different random seeds. Forecasting horizon $H \in \{24, 36, 48, 60\}$ for ILI dataset and $H \in \{96, 192, 336, 720\}$ for the other datasets.

as 300k to as high as to 900M. We display the number of parameters of the models used in Table 1 as a histogram in Figure. 7. This variance in parameters counts is much larger than that of baseline model, PatchTST, which spans from 900k to 4M. Note that the majority of the model parameters of LGPred stem from the fully connected layers within $f_{feat}$ and $g_W$ of the predictor generator. Taking the models used in Table 1 again as an example, these layers account for an average 94.37% of the parameters. For the broad application of LGPred across diverse scenarios, it is important to reduce the complexity of LGPred by refining the structure of the predictor generator. However, we leave this problem for future works.

## B.8 Architecture of Representation Module

In the main paper, we utilize the mixer-like fully connected layer for the trend representation module, and dilated temporal convolution layer for the seasonality representation module. In this section, we test various architectures for trend and seasonality representation modules and compare the performance with our original choices. We display in Table 12 the experiment results with various architectures for the representation modules using ECL and Exchange datasets. In Table 12, FC and CNN denote the mixer-like fully connected layer and dilated temporal convolution layer that we adopt in main paper respectively. RNN and Attn mean the standard LSTM layer and Transformer layer each. The experiment results indicate that our original combination utilizing FC layer for trend representation and CNN for seasonality representation shows the best performance. The results in Exchange dataset also confirm that our assumption that RNN, FC, and transformer layer is more suitable for capturing the trend characteristics compared to CNN is correct.

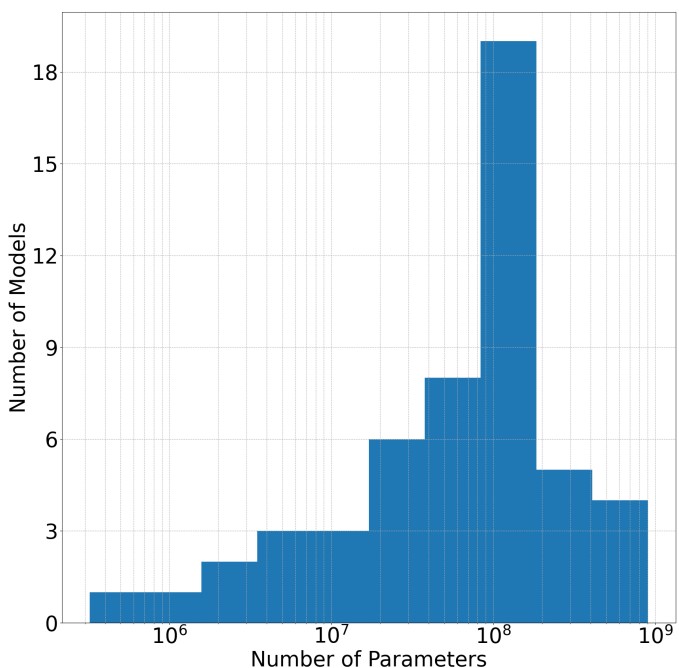

Figure 7: Histogram of the number of trainable parameters of the models used in Table 1.

| Trend | RNN | | | | CNN | | | | FC | | | | Attn | | | |
|---|---|---|---|---|---|---|---|---|---|---|---|---|---|---|---|---|
| Seasonal | RNN | CNN | FC | Attn | RNN | CNN | FC | Attn | RNN | CNN | FC | Attn | RNN | CNN | FC | Attn |
| Electricity 96 | 0.132 | 0.131 | 0.133 | 0.132 | 0.133 | 0.132 | 0.132 | 0.131 | 0.134 | **0.130** | 0.139 | 0.136 | 0.133 | 0.133 | 0.133 | 0.132 |
| Electricity 192 | 0.145 | **0.144** | **0.144** | **0.144** | 0.146 | 0.147 | 0.147 | 0.147 | 0.151 | **0.144** | 0.156 | 0.154 | 0.147 | 0.156 | 0.148 | **0.144** |
| Electricity 336 | 0.161 | 0.161 | 0.162 | 0.159 | 0.163 | 0.162 | 0.163 | 0.162 | 0.166 | **0.158** | 0.171 | 0.163 | 0.163 | 0.162 | 0.164 | 0.164 |
| Electricity 720 | 0.204 | 0.199 | 0.204 | 0.201 | 0.198 | 0.202 | 0.202 | 0.205 | 0.197 | **0.190** | 0.201 | 0.198 | 0.200 | 0.200 | 0.200 | 0.202 |
| Exchange 96 | 0.092 | 0.091 | 0.091 | 0.091 | 0.094 | 0.094 | 0.094 | 0.094 | **0.079** | **0.079** | **0.079** | **0.079** | 0.096 | 0.096 | 0.096 | 0.096 |
| Exchange 192 | 0.174 | 0.283 | 0.175 | 0.203 | 0.312 | 0.336 | 0.281 | 0.441 | 0.192 | **0.152** | 0.218 | 0.255 | 0.272 | 0.740 | 0.400 | 0.236 |
| Exchange 336 | 0.516 | 0.556 | 0.496 | 0.527 | 0.811 | 0.993 | 0.810 | 0.847 | 0.279 | **0.261** | 0.272 | 0.297 | 0.881 | 0.858 | 0.743 | 0.773 |
| Exchange 720 | 0.800 | 0.670 | 0.489 | 0.687 | 6.306 | 8.324 | 8.000 | 3.376 | 0.528 | **0.422** | 0.492 | 0.537 | 0.973 | 0.907 | 0.589 | 1.053 |

Table 12: Multivariate long-term forecasting MSE with various network architecture for representation module. The **best results** are highlighted in **bold**.

## C  DETAILED PROCEDURE OF LGPRED

In this section, we provide more detailed explanation on the prediction procedure of LGPred with a pseudo-code presented in Algorithm 1. `TimeSeriesDecomposition` denotes the decomposition module decomposing time series into trend $\mathcal{T}$ and seasonality $\mathcal{S}$ using the moving average kernel (line 1). The representation modules $f_{\mathcal{T}}$ and $f_{\mathcal{S}}$ extracts representation from trend $\mathcal{T}$ and seasonality $\mathcal{S}$ (line 2-3), and the fully-connected layers $f_{feat}^{\mathcal{T}}$ and $f_{feat}^{\mathcal{S}}$ compress the representations into features $h_{\mathcal{T}}$ and $h_{\mathcal{S}}$ (line 4-5). Using the weight generators $[g_W^{\mathcal{T}}, g_W^{\mathcal{S}}]$ and bias generators $[g_b^{\mathcal{T}}, g_b^{\mathcal{S}}]$, LGPred generates weight $W_{gen}$ by adding trend weight and seasonality weight and bias $b_{gen}$ by adding trend and seasonality biases (line 6-11). The prediction for the input time series is then conducted based on the generated weight and bias. First, the input is normalized by subtracting the last value $\mathbf{x}_{t-1}$ from the input $\mathbf{X}_{t-L:t}$ (line 12). Using down-project template weight $W_{down}$, the normalized input $\mathbf{X}_{norm}$ is projected into latent feature $\mathbf{X}_{latent}$ (line 13). Then, we multiply the generated weight $W_{gen}$ to $\mathbf{X}_{latent}$ (line 14), and then multiply product $\hat{\mathbf{X}}_{latent}$ with the up-project template weight $W_{up}$ to get the prediction $\hat{\mathbf{X}}_{up}$ with length $H$ (line 15). The generated bias and template bias are

added to the $\hat{\mathbf{X}}_{up}$ (line 16), and the final prediction $\hat{\mathbf{X}}_{t:t+H}$ is obtained by denormalizing $\hat{\mathbf{X}}_{up}$ by adding the last value of input $\mathbf{x}_{t-1}$ to $\hat{\mathbf{X}}_{up}$ (line 17).

---

**Algorithm 1** Prediction Procedure of LGPred

---

    **Input** : $\mathbf{X}_{t-L:t} = [\mathbf{x}_{t-L}, \cdots, \mathbf{x}_{t-1}] \in \mathbb{R}^{L \times m}$ where $L$ is the look-back window size

    **Output** : $\hat{\mathbf{X}}_{t:t+H} = [\hat{\mathbf{x}}_t, \cdots, \hat{\mathbf{x}}_{t+H-1}] \in \mathbb{R}^{H \times m}$ where $H$ is the length of forecasting horizon.

1: $\mathcal{T}, \mathcal{S} \leftarrow \texttt{TimeSeriesDecomposition}(\mathbf{X}_{t-L:t})$

2: $\mathcal{H}_\mathcal{T} \leftarrow f_\mathcal{T}(\mathcal{T})$                                            ▷ Extract Representation

3: $\mathcal{H}_\mathcal{S} \leftarrow f_\mathcal{S}(\mathcal{S})$

4: $h_\mathcal{T} \leftarrow f_{feat}^\mathcal{T}(\mathcal{H}_\mathcal{T})$                              ▷ Compress Representation into Feature

5: $h_\mathcal{S} \leftarrow f_{feat}^\mathcal{S}(\mathcal{H}_\mathcal{S})$

6: $W_{gen}^\mathcal{T} \leftarrow g_W^\mathcal{T}(h_\mathcal{T})$                                      ▷ Predictor Generation Start

7: $b_{gen}^\mathcal{T} \leftarrow g_b^\mathcal{T}(h_\mathcal{T})$

8: $W_{gen}^\mathcal{S} \leftarrow g_W^\mathcal{S}(h_\mathcal{S})$

9: $b_{gen}^\mathcal{S} \leftarrow g_b^\mathcal{S}(h_\mathcal{S})$

10: $W_{gen} \leftarrow W_{gen}^\mathcal{T} + W_{gen}^\mathcal{S}$

11: $b_{gen} \leftarrow b_{gen}^\mathcal{T} + b_{gen}^\mathcal{S}$                              ▷ Predictor Generation End

12: $\mathbf{X}_{norm} \leftarrow \texttt{Normalize}(\mathbf{X}_{t-L:t})$                       ▷ Prediction Start

13: $\mathbf{X}_{latent} \leftarrow W_{down} * \mathbf{X}_{norm}$

14: $\hat{\mathbf{X}}_{latent} \leftarrow W_{gen} * \mathbf{X}_{latent}$

15: $\hat{\mathbf{X}}_{up} \leftarrow W_{up} * \hat{\mathbf{X}}_{latent}$

16: $\hat{\mathbf{X}}_{up} \leftarrow \hat{\mathbf{X}}_{up} + b_{gen} + b$

17: $\hat{\mathbf{X}}_{t:t+H} \leftarrow \texttt{DeNormalize}(\hat{\mathbf{X}}_{up})$                         ▷ Prediction End

---

