# OpenReview forum: "Learning to Generate Predictor for Long-Term Time Series Forecasting"
_ICLR.cc/2024/Conference — Submitted to ICLR 2024_

### Official Review · Reviewer_qLBd · 2023-10-29

**Soundness:** 3 good
**Presentation:** 3 good
**Contribution:** 3 good
**Rating:** 6
**Confidence:** 4

**Summary:**

The paper proposes a new framework called Learning to Generate Predictor (LGPred) for long-term time series forecasting. The key idea is to generate a linear predictor dynamically tailored to the input time series, to overcome limitations of fixed linear predictors. LGPred uses time series decomposition into trend and seasonality components. Separate representation modules extract features from each component. A predictor generator uses the extracted features to generate the weights and biases of a linear predictor suited to the input series. A template predictor with bottleneck architecture is used to incorporate common forecasting knowledge and reduce computation cost. Experiments show state-of-the-art performance on 6 benchmark datasets covering disease, economics, energy, traffic and weather domains.

**Strengths:**

(1) Novel idea of generating parameters of predictor based on input series, enabling adaptation to each series.

(2) Bottleneck template predictor shares knowledge among different time series and reduces computational cost.

(3) Well-written paper and easy to understand.

**Weaknesses:**

(1) The proposed method includes multiple modules and each of them have their own hyper-parameters to tune. Extensive hyper-parameter tuning was not favorable. Also, I was not clear why a bottleneck architecture was used for template predictor and intuition behind this design was not discussed.

(2) Time series decomposition is a standard technique already used by some prior works.

(3) Experimental results do not include mean value and standard deviations. It will be good to know if the proposed method is sensitive to initialization.

(4) Results are mainly conducted on time series datasets with simple patterns. What if the patterns are complicated and hard to capture for linear predictors?  More experimental results or at least discussion of limitations are needed. Time series can vary significantly in terms of distribution and it is good to know when an algorithm can perform good/bad.

**Questions:**

See my comments in Weaknesses

---

> ### Author Response · Authors · 2023-11-22
> **Official Comment to Reviewer qLBd**
>
> We thank the reviewer for the positive comments and valuable feedback. Concerning the questions and concerns that the reviewer raised, we provide our response below.
>
> - Hyperparameter
>     - For the hyperparameter sensitivity analysis, we added the results in appendix B6. Since our LGPred is a network that generates the network, the sensitivity to the hyperparameters related to the predictor generation is somewhat inevitable. However, we believe that our methodology of generating the predictor adaptively to the given time series has clear value even considering this limitation.
> - Bottleneck architecture for template predictor
>     - The reason we utilize the bottleneck architecture for our template predictor is to reduce the number of parameters to generate. As we mentioned in page 5, the number of parameters in predictor generator is significantly reduced due to the bottleneck architecture, although the number is still large. Also, the intuition behind the bottleneck architecture is the fact that the rank required for the linear predictor to predict time series is low. We conduct an experiment lowering the rank of the linear predictor by dividing the linear predictor into a bottleneck architecture of two linear layers having small latent dimension and measuring the performance. Our experiment results show that the performance is preserved even though the rank is much lower than the input and output length (d_latent =32). While subsequent research revealed that a larger latent dimension (rank) is required for predictor generation in most cases, the effectiveness of the bottleneck architecture in linear predictors and the effect of reducing the complexity of the bottleneck architecture are still meaningful.
> - Time series decomposition and our contribution.
>     - As the reviewer correctly stated, the time series decomposition is a standard technique and we do not claim it to be our contribution. However, our contribution lies in utilizing the time series decomposition for analyzing and understanding the characteristics of time series instead of directly utilizing it for time-series forecasting.
> - Limitation of utilizing linear predictor.
>     - In the paper, we conduct experiments on well-known benchmark datasets, which contain relatively simple patterns that can be predicted with a linear predictor. If the patterns become more complicated and harder, both linear predictor and our LGPred may fail to predict the patterns. However, this point holds true not only for linear predictors but also for more complicated models, and at least for the benchmarks covered so far, the linear predictor works as well as the other models. Therefore, we focus on other limitations of linear predictors rather than the pattern prediction capability. We focus on the limitation that the linear predictors cannot make predictions reflecting the dynamically changing characteristics of time series. Our experiment with synthetic data proves that this limitation exists, and we can overcome this limitation using our LGPred. Another limitation of the linear predictor is that it cannot utilize theㅅ cross-variate correlation in multivariate time-series data. This limitation remains in our LGPred, and we already discussed in the Discussion section. Furthermore, we believe the limitation of linear predictor can be overcome by applying our LGPred to other time series forecasting models, which is capable by LGPred's orthogonality to other forecasting model. Our experiment results in appendix B.2. show the possibility of combining LGPred with the existing forecasting model and achieving better performance.

---

### Official Review · Reviewer_LX8c · 2023-10-31

**Soundness:** 2 fair
**Presentation:** 3 good
**Contribution:** 2 fair
**Rating:** 5
**Confidence:** 5

**Summary:**

This paper proposes a model (LGPred) which learns the predictor for each sample in the long-term time series forecasting tasks. The LGPred generates a part of weights and bias for the projection from the input to the output, based on representations learnt from the trend and seasonality of each sample. Experiments on several benchmark datasets are conducted to evaluate the effectiveness of LGPred.

**Strengths:**

The proposed LGPred can generate dynamic predictors for different samples, which is novel for the long-term time series prediction tasks. The paper is well-written in general and easy to understand.

**Weaknesses:**

1. Some parts in the Preliminary and Method sections are not clear, e.g.,

-It is not clear why time series forecasting with T>48 is considered as LTSF problem, are there any reasons or references?

-Why change the number of channels in the trend block?

-The dilated temporal convolutional network should be introduced, in case some readers do not have related background Knowledge.

2. The comparison with baselines may be unfair and experiments are insufficient.

-I think it is unfair to compare with PatchTST/64 which uses lookback window length 512 only. As shown in the Figure 2 of the PatchTST paper, the performance is changed with different lookback windows. It is better to choose the best results from different lookback windows for PatchTST for a fair comparison. In addition, even based on the current results shown in the Table 1, the proposed LGPred cannot beat PatchTST/64.

-It is better to add the results of PatchTST in Table 3 due to its superiority.

-There is no complexity analysis between the proposal and baselines.

-It is better to provide some experiment results of using RNN and transformer for the trend component.

**Questions:**

Same to the Weaknesses.

---

> ### Author Response · Authors · 2023-11-22
> **Official Comment to Reviewer LX8c**
>
> We appreciate the reviewer providing valuable comments for our paper. For the concerns and issues raised by reviewer, we provide our responses below.
>
> - Reference of T>48 is considered as LTSF problem.
>     - Considering the definition of LTSF problem, the first work to distinguish LTSF problem from regular TSF problem is Informer [1], and they indicated that predicting 48 points or less as short-term forecasting and show that the forecasting performance of old methods start to decrease sharply around T=48. We follow the setting of the informer.
> - Reason why the number of channels in Trend block in Fig. 2
>     - During feature extraction, we decided to maintain the length of time dimension in the feature, to preserve the temporal information. However, for the channel dimension, since our generated predictor is applied in channel independence manner, we chose to mix the channel information and encode it into the features with length of d_rep instead of preserving it. This is the reason why we change the number of channels in the trend block of Fig.2, as explained in page 4.
> - Explanation about dilated temporal convolution network
>     - Regarding the dilated temporal convolution network, we added some explanation about dilated TCN and why we utilize dilated TCN for seasonality representation module in page 4.
> - Comparison with PatchTST
>     - In Table 1 and 2, we update the PatchTST performance by choosing the model showing the better MSE score between PatchTST/64 and PatchTST/42. Note that there are only a few experiments in which the best model changed due to this update.
>     - For the comparison between PatchTST and LGPred, we believe that our LGPred shows at least the same level of performance as PatchTST. For example, in Table 1, our LGPred shows equal or higher performance than PatchTST in 16 out of 32 experiments for MSE, 12 out of 32 experiments for MAE. In Table 2, our LGPred clearly outperforms PatchTST in most cases.
>     - In Table 3, we did not include the result of PatchTST in our first submission. We obtained the scores of baseline models in Table 3 from the TimesNet paper [2], and the performance of PatchTST is not included in the original paper. Note that the results in short input length situations have not been reported in the PatchTST paper [3]. However, we ran additional experiments using the implementation in Time-Series-Library (https://github.com/thuml/Time-Series-Library), which is an official implementation of TimesNet, and added the results in Table 3. The result shows that LGPred outperforms PatchTST when the input length is limited in most cases.
> - Complexity analysis
>     - For the complexity analysis, we added the results in appendix B.7. Thank you for your comment and please check section B.7 in appendix of our updated paper.
> - Using RNN and Transformer for the trend representation module
>     - In appendix, we added the experiment results using the RNN and transformer architecture for trend representation module. The experiment results indicate that utilizing FC layer for trend component and CNN for seasonality component shows the best performance. Furthermore, results also confirm that our assumption that RNN, FC, and transformer layer is more suitable for capturing the trend characteristics compared to CNN is correct.
>
> [1] Haoyi Zhou, et al. Informer: Beyond efficient transformer for long sequence time-series forecasting. In Proceedings of the AAAI conference on artificial intelligence, volume 35, pp. 11106–11115, 2021
>
> [2] Haixu Wu, etal. Timesnet: Temporal 2d-variation modeling for general time series analysis. In The Eleventh International Conference on Learning Representations, 2022
>
> [3] Yuqi Nie, et al. A time series is worth 64 words: Long-term forecasting with transformers. In The Eleventh International Conference on Learning Representations, 2022.

---

> > ### Comment · Reviewer_LX8c · 2023-12-05
> >
> > Thank authors for the response. Considering the performance and complexity of the proposal, I would like to maintain my rating.

---

### Official Review · Reviewer_qCaW · 2023-11-01

**Soundness:** 1 poor
**Presentation:** 2 fair
**Contribution:** 1 poor
**Rating:** 5
**Confidence:** 3

**Summary:**

This paper proposed a learning-to-generate-predictor model, LGPred, for long-term forecasting. In particular, LGPred consists of two parts, a weights generator, and a feature extraction, and then uses a bilinear-type structure to merge them. Moreover, the seasonality trend decomposition is used in the weights generator. Numerical results on 9 datasets are reported.

**Strengths:**

The usage of bilinear structure seems new in the time series forecasting domain.

**Weaknesses:**

1. The term *Learning to Generate Predictor* is a little bit overstated from my perspective. My first impression would be a meta-learning model is considered. However, after reading the paper and codes. It seems just a usage of bilinear-type layer for me. I appreciate applying the bilinear layer since it seems not being used in recent forecasting literature. But *Learning to Generate Predictor* may not be the best term to summarize the model novelty for me. If the authors still prefer using *Learning to Generate Predictor*, it would be better to add more discussion to clearly state the difference from the meta-learning type model.

2. The statement "*LGPred is the first attempt at adaptively generating a predictor reflecting the characteristics of each time series.*" seems also a little bit overclaimed. For example, in DeepAR, the network will first generate the $\mu$/$\sigma$ or $\mu$/$\alpha$ for Gaussian distribution or negative binomial (NB) distribution respectively. During the inference stage, the forecasting point will be sampled from the Gaussian/NB distribution. In this case, the Gaussian/NB distribution can be viewed as the *Predictor*, and the parameters in the predictor are learned with a network.

3. The test data loader sets `drop_last = True`. In this case, the last several test samples are ignored, which will impact the accuracy of results in Table 1- Table 3. It would be better if the authors could fix it.

4. It seems that the random control experiments are not conducted.


Reference:

DeepAR: Salinas, David, Valentin Flunkert, Jan Gasthaus, and Tim Januschowski. "DeepAR: Probabilistic forecasting with autoregressive recurrent networks." International Journal of Forecasting 36, no. 3 (2020): 1181-1191.

**Questions:**

1. The main results in Table 1 - Table 3 are from the model after hyperparameter searching. I'm wondering if the authors can provide a sensitive analysis of the parameter choices to further highlight the robustness of the proposed model.

---

> ### Author Response · Authors · 2023-11-22
> **Official Comment to Reviewer qCaW**
>
> We appreciate the reviewer’s thoughtful comments. Below, we reply to the comments raised by the reviewer.
>
> - Relation between meta-learning type model and LGPred
>     - Reviewer mentioned that the LGPred is a bilinear type layer rather than a meta-learning type model. We respectfully disagree. In general, meta-learning type models adopt a two-tier learning; the inner loop that conducts the adaptation to the given task and the outer loop that trains the model to better adapt to the given task. For our LGPred, the predictor generation path works as the inner loop conducting the adaptation to the given time series, and the training of the model works as the outer loop. In this sense, the work of LGPred is a kind of meta-learning type framework, rather than a bilinear structure. We draw inspiration from meta-learning model, and we make this point clearer in the introduction section of our updated paper. While applying the predictor originated from the input time-series to the input time-series bears some resemblance to bilinear structure, we believe the term ‘learning to generate predictor’ better explains the novelty of our LGPred.
> - Learning to generate in prior work DeepAR
>     - In DeepAR, the mean and standard deviation of prior distribution is predicted, and the prediction values are probabilistically generated from the distribution. However, considering the fact that the objective of DeepAR is probabilistic forecasting, the mean and standard deviation are the prediction output itself rather than the predictor. In this sense, we believe that our statement “LGPred is the first attempt at adaptively generating a predictor reflecting the characteristics of each time series” is not overclaimed.
> - ‘drop_last’ config in test loader
>     - Before answering, we are very impressed that the reviewer kindly checked out our code. The reviewer kindly informed that the drop_last config is set to be True, and this results in some test samples being ignored. We agree with the reviewer’s point. However, all prior works evaluate their methods under the same situation, and we follow the experimental setup for fair comparison. Please see the following official implementation provided by the authors.
>         - Informer: https://github.com/zhouhaoyi/Informer2020/blob/main/exp/exp_informer.py (L77)
>         - Autoformer: https://github.com/thuml/Autoformer/commit/d9100709b04e3e8361170794eba4f47b1afb217f (L20, committed after publish)
>         - FEDformer: https://github.com/MAZiqing/FEDformer/blob/master/data_provider/data_factory.py (L19)
>         - TimesNet: https://github.com/thuml/Time-Series-Library/blob/main/data_provider/data_factory.py (L28)
>         - PatchTST: https://github.com/yuqinie98/PatchTST/blob/main/PatchTST_supervised/data_provider/data_factory.py (L19)
>         - ETSformer: https://github.com/salesforce/ETSformer/blob/main/data_provider/data_factory.py (L19)
>         - LTSF-Linear: https://github.com/cure-lab/LTSF-Linear/commit/6fe4c28ff36b4228792f2bbe513e807577e4a57e (L20, committed after publish)
> - Random control and Hyperparameter sensitivity analysis
>     - Thank you for the reviewer’s kind comment to improve the completeness of our paper. For the random control experiment and hyperparameter sensitivity analysis, we added the results in appendix B5 and B6. Note that some of our original hyperparameter settings in Table 1 turn out to be vulnerable to random initialization, and we updated some scores in Table 1. Regarding the experiments in Table 3, we will update the random control experiment results if accepted.

---

### Official Review · Reviewer_dX6j · 2023-11-01

**Soundness:** 2 fair
**Presentation:** 3 good
**Contribution:** 2 fair
**Rating:** 3
**Confidence:** 4

**Summary:**

This paper proposes the Learning to Generate Predictor (LGPred) framework, a novel approach to enhancing linear time series forecasting models. LGPred adaptively generates a linear predictor tailored to the specific characteristics of a given time series by time series decomposition. This allows the model to discern and adapt to each time series' unique trend and seasonality components. Experimental evidence presented in the paper indicates that LGPred consistently delivers top-tier performance on various benchmarks.

**Strengths:**

1. The proposed method provides clear motivation for its designs.
2. Empirical results showcase commendable performance, effectively outperforming many preceding methodologies.

**Weaknesses:**

1. The paper seems to omit discussions on contemporary related works. Notably, the structure of the proposed trend representation module is almost the same as TSMixer [1]. Furthermore, TiDE [2] has previously delved into refining linear models specifically for time series forecasting. Given the architectural similarities between these MLP-based models, a more in-depth comparison and differentiation would enhance clarity.
2. The absence of comprehensive ablation studies leaves the intrinsic value of each component in the proposed method ambiguous. For instance, the ablation analysis in [1] revealed that simpler stacked linear models (i.e., TMix-Only) could rival the performance of the presented methodology. This raises questions regarding the neccesity of LGPred's individual components.
3. The delineation of the dimensions for the linear and fully-connected layers remains ambiguous. For multivariate time series data, these layers could be applied across either time or feature dimensions, as depicted in Figure 2. Unfortunately, the descriptions on the predictor generator and template predictor (page 4) do not explain the dimensional characteristics of these layers adequately.

[1] Chen, Si-An, et al. "TSMixer: An All-MLP Architecture for Time Series Forecasting." Transactions on Machine Learning Research. 2023

[2] Das, Abhimanyu, et al. "Long-term Forecasting with TiDE: Time-series Dense Encoder." Transactions on Machine Learning Research. 2023

**Questions:**

1. Does the proposed architecture incorporate any non-linear activation functions? It's worth noting that certain linear modules, such as $b_{gen}$, might be redundant given that the concatenation of multiple linear layers essentially functions as a single linear layer.
2. Considering the insights from recent works ([1], [2]) highlighting the potential inadequacy of LTSF benchmarks in reflecting models' capability in handling cross-variate correlations, can the LGPred framework be generalized to tackle more intricate datasets like M5 or Favorita, as explored in [1] and [2]?

---

> ### Author Response · Authors · 2023-11-22
> **Official Comment to Reviewer dX6j**
>
> We appreciate the reviewer's valuable comments. For the issues and concerns raised by the reviewer, we provide our responses below.
>
> - Comparison with TSMixer and TiDE
>     - We appreciate the reviewer suggesting the interesting related works. However, we believe the contribution of TSMixer[1] and TiDE[2] is clearly different from our LGPred. These works mainly focus on utilizing the multivariate features, while our LGPred focuses on adapting a model to a given time series. Meanwhile, the architectural similarity between these TSMixer and the trend representation module of our LGPred is an interesting point, and we added the comparison in method and related work section. Regarding the performance comparison, we wish the reviewer to understand the performance comparison is not conducted in the paper since these works are concurrent with our work published in August and September.
> - Ablation studies to show the value of each component
>     - We already have presented the ablation study results in appendix B.1. We believe that the results in appendix B1 prove the value and necessity of the components of our LGPred.
> - Dimensional characteristics of FC layers in LGPred
>     - We want to note that our LGPred conducts forecasting in channel-independent manner, i.e., the same predictor is applied across all features, as done in prior works [3][4]. While the architectures and layers for multivariate features such as mixer architecture or temporal convolutional layer are applied in predictor generation path, the objective of these architecture is to generate the predictor suitable for prediction of all features, not utilizing multivariate correlation in prediction. To this end, as we clearly stated in predictor generator section in page 4, we flatten the feature and use it to generate predictor shared across all channels.
> - Nonlinear activation function
>     - As reviewer clearly pointed out, the concatenation of multiple linear layers functions as a single linear, and this is just what we intended. While linear predictor is proven to be effective in long-term forecasting task in a prior work [3], we believe that the limitation of linear predictor is that its simplistic architecture cannot reflect the dynamically changing characteristics of time series. To address this limitation, our goal is to adapt a ‘linear’ predictor suitable for given time series, and the final predictor model is a linear predictor without any nonlinear activation. When we apply our LGPred to forecasting models with nonlinearity as done in Appendix B.2, forecasting model in prediction path also utilizes nonlinearity. Note that we utilize the non-linear activation functions of GeLU in the predictor generation path, as we already have stated in Section 3 and Figure 2.
> - Extension to datasets with cross-variate correlation
>     - Since our base model is a linear predictor, which is a channel independent model, handling cross-variate correlation with LGPred is not straightforward. However, our LGPred is an orthoginal method with other time series forecasting method and we show that it is possible to combine LGPred with other models such as PatchTST and TimesNet in appendix B.2, and we believe that by combining with multivariate forecasting models such as TSMixer and TiDE, LGPred framework can tackle the multivariate forecasting tasks such as M5 or Favorita. Note that we already stated this limitation and possibility in the discussion section.
>
> [1] Chen, Si-An, et al. "TSMixer: An All-MLP Architecture for Time Series Forecasting." Transactions on Machine Learning Research. 2023
>
> [2] Das, Abhimanyu, et al. "Long-term Forecasting with TiDE: Time-series Dense Encoder." Transactions on Machine Learning Research. 2023
>
> [3] Ailing Zeng, et al. Are transformers effective for time series forecasting? In Proceedings of the AAAI conference on artificial intelligence, volume 37, pp. 11121–11128, 2023
>
> [4] Yuqi Nie, et al. A time series is worth 64 words: Long-term forecasting with transformers. In The Eleventh International Conference on Learning Representations, 2022.

---

### Meta-Review · Area_Chair_3dHF · 2023-12-07

**Metareview:**

This paper proposes a new LGPred forecasting model for LTSF that uses a two stage approach involving a seasonality and trend representation using time series decomposition, followed by generating weights and biases for a linear predictor. The paper benchmarks their model on standard LTSF datasets.  Reviewers raised concerns around whether this is truly a meta-learning approach as opposed to really being a two stage architecture built upon previous works (D-Linear, Dilated TCN and MLPs in time series, and time series decomposition). There were also concerns raised (which AC agrees with) on the empirical results, since the model does not really outperform the state-of-the-art (patchTST), despite the complexity of the proposed architecture. I would urge the authors to consider strengthening the paper with better presentation, more competitive empirical performance and extending the evaluation to more complex LTSF datasets like M5 and Favorita.

**Justification For Why Not Higher Score:**

As two of the reviewers pointed out, the paper has two major flaws - first, it suffers from overclaiming on the learning-to-predict front, when this really seems like a two stage architecture built upon previous works (D-Linear, Dilated TCN and MLPs in time series, and time series decomposition). More importantly, given the complexity of the proposed architecture, the empirical results do not really improve over the existing state-of-the-art PatchTST approach.

**Justification For Why Not Lower Score:**

N/A

---

### Decision · Program_Chairs · 2024-01-16

Reject